# On the Genesis of a Catalyst: A Brief Review with an Experimental Case Study

Simón Yunes [1,*], Jeffrey Kenvin [1] and Antonio Gil [2,*]

1 Micromeritics Instrument Corporation, 4356 Communications Drive, Norcross, GA 30093, USA; jeffrey.kenvin@micromeritics.com
2 INAMAT2, Science Department, Public University of Navarra, 31006 Pamplona, Spain
* Correspondence: simon.yunes@micromeritics.com (S.Y.); andoni@unavarra.es (A.G.)

**Abstract:** The science of catalysis has a direct impact on the world economy and the energy environment that positively affects the environmental ecosystem of our universe. Any catalyst, before being tested in a reaction, must undergo a specific characterization protocol to simulate its behavior under reaction conditions. In this work, these steps that must be carried out are presented, both generically and with examples, to the support and to the catalyst itself before and after the reaction. The first stage consists of knowing the textural and structural properties of the support used for the preparation of the catalysts. The specific surface area and the pore volume are fundamental properties, measured by $N_2$ adsorption at $-196\ °C$ when preparing the catalyst, dispersing the active phase, and allowing the diffusion and reaction of the reactants and products on its surface. If knowing the structure of the catalyst is important to control its behavior against a reaction, being able to analyze the catalyst used under the reaction conditions is essential to have knowledge about what has happened inside the catalytic reactor. The most common characterization techniques in heterogeneous catalysis laboratories are those described in this work. As an application example, the catalytic conversion of $CO_2$ to $CH_4$ has been selected and summarized in this work. In this case, the synthesis and characterization of Cu and Ni catalysts supported on two $Al_2O_3$ with different textural properties, 92 and 310 $m^2/g$, that allow for obtaining various metallic dispersions, between 3.3 and 25.5%, is described. The catalytic behavior of these materials is evaluated from the $CO_2$ methanation reaction, as well as their stability from the properties they present before and after the reaction.

**Keywords:** catalyst; catalytic support; metal oxide; textural properties; structural properties; characterization techniques; chemical characterization; physisorption; chemisorption; metal dispersion; catalytic activity; catalytic deactivation; in situ characterization





## 1. Introduction

The word catalyst is one of the most repeated words in the literature and it is in direct relation to many processes for several applications such as refining, pharmacy, mining, painting, and many others. Catalysts play an important role in converting species into more beneficial products that make a large contribution to the overall world economy, especially processes that convert $CO_2$ into more beneficial hydrocarbon molecules [1–17]. Catalysis, however, is a process in which a catalyst is involved in a reaction to proceed at a faster rate or under various conditions than otherwise possible, while not being consumed by the reaction.

Several steps are followed to synthesize a heterogeneous catalyst: selection of the support, followed by selection of the oxides that will become the active species, and, finally, synthesis of the catalyst according to one of the well-known methodologies [18,19]. Upon proper characterization of the support, followed by the preparation of the catalyst, testing becomes the critical point to produce the expected products and to have a long-lasting catalyst. Proper tools are required for this study that include in situ characterization of the

catalyst, especially after deactivation to elucidate and understand the causes of deactivation and, finally, to produce a catalyst that lasts longer under reaction conditions and produces the best activity and selectivity for any specific reaction.

There are several possible ways to limit the release of greenhouse gases into the environment. The first step is to limit the use of fossil fuels by adopting the practical use of green energy sources, which minimizes gas emissions. Simultaneously, technologies for carbon capture and storage (CCS) and carbon capture and utilization (CCU) should be applied. While CCS processes end with the storage of $CO_2$, which does not solve the main issue, CCU processes involve capturing $CO_2$ and converting it into valuable products [20]. This storage and conversion can be performed using a variety of processes. Absorption into liquid is a suitable storage strategy for $CO_2$, but it is limited by the high energy demands of regenerating the solvent. Adsorption on solids is affected by temperature and pressure and is a current process to generate innovative porous materials [21]. The membrane technique has a variety of advantages over other CCS technologies, such as applicability in isolated areas, simplicity of maintenance, a low-cost installation, and fewer chemical and energy requirements [22,23]. Photocatalytic reduction is initiated through direct sunlight, which is considered a renewable energy source, while electrochemical and plasma technology relies on electricity. Additionally, chemical storage can be implemented on the current line of infrastructure with a high capability [22]. The advanced technology of water electrolysis (power to gas, P2G) has contributed convenient enhancements in carbon dioxide hydrogenation as a source of green fuel [24].

Significant improvements have been achieved in transforming $CO_2$ into valuable single-carbon materials, such as carbon monoxide, methane, methanol, and formic acid, among others. The reaction of the reverse water gas shift (RWGS) produces carbon monoxide, while heterogeneous catalysis has been utilized to obtain methanol [25]. Thermodynamically, the methanation of $CO_2$ and CO is quicker than that of other reactions for hydrocarbon production. $CO_2$ gas is highly stable and molecule separation is costly [26]. This stability has led to low production due to the low adsorption rate of $CO_2$ in the catalysts. A full understanding of the complexity of the mechanism of $CO_2$ conversion into hydrogen fuel requires gaining more information, such as from studying the micro mechanisms of the Sabatier reaction [27]. Compounds with small carbonylicity are more reactive during the addition reaction in comparison with the high carbonylicity compounds [28]. $CO_2$ conversion to methane requires efficient catalysts to be successful. Recently, this has been the aim of much research that has attempted using various catalyst designs [29]. Metal–support catalysts are extensively used for $CO_2$ fixation reactions with a variety of metals and catalyst supports. Although several transition metals are suitable for the methanation reaction, supported Co, Cu, Fe, and Ni catalysts are extensively used for $CO_2$ methanation due to their high catalytic performance and cost-effective nature [30], but metal catalysts lose activity quickly during the methanation reaction due to their carbon deposition [29]. The roles of the support, metal loading, the preparation method, and additives have been investigated to enhance the performance of the catalysts. According to the literature, the support might play a role in enhancing metal dispersion and tuning the structure of the catalyst surface, which could improve $CO_2$ adsorption and affect the reaction mechanism [30]. Alumina, ceria, magnesia, silica, titania, and zirconia are well-studied support materials that are thermally stable and provide significantly high surface areas [31]. They increase the stability of metal-based catalysts by improving their interaction with the active metals [32]. The acidity/basicity of the supports weakens the interaction with $CO_2$. To improve the interaction, the basicity of the metal catalysts is enhanced with the presence of promoters [33]. Finally, it should not be forgotten that other types of materials such as metal nitrides, hydrides, and carbides could also intervene in the CCS and CCU processes. However, for simplicity, this work has been reduced only to the case of metal oxides and metals supported.

This review work presents the most important steps to produce a catalyst: the properties that determine its performance and what experimental methods available in a large

number of academic and business laboratories can be used. Finally, a practical example of the synthesis and characterization of catalysts applied in the methanation of $CO_2$ is addressed. Special relevance is made to the characterization of the catalyst through techniques that allow its characterization under conditions very similar to those in which the catalytic reaction is carried out. In particular, from the use of the named In Situ Catalyst Characterization System (ICCS).

## 2. Genesis of a Catalyst

### 2.1. Selection of the Support

This is the first and most important task to be considered when preparing a catalyst that will later be designated for a certain reaction. In general, they do not have to present specific characteristics for a reaction, their surface properties must allow the dispersion of the active metallic phase. Synthesizing a catalyst is normally a sophisticated and gentle art and is based on the selection of solid support made of refractory oxides (alumina, ceria, silica, titania, zirconia, carbon, and many others) and an active phase, each containing their own physical and chemical properties, which can evolve during preparation. First, the selected support must fulfill certain requirements:

(a) It should withstand high temperature and pressure with minimal sintering;
(b) It should have an adequate texture; that is, sufficient surface area to lodge the necessary oxide particles that will play the role of the active species. It should also have adequately sized pores to facilitate diffusion of the reactants through the pores and the ability for the product to diffuse out of the pore [34,35];
(c) It must have a certain surface acidity/basicity when required by the reaction;
(d) Other interesting characteristics must be sometimes required such as morphology or any structure arrangement that yields to stabilize the active species.

### 2.2. Selection of the Metal Active Sites

Metal oxides serve as active elements on the catalyst. This second step in this study is also very important and it must be carefully considered in the synthesis of a catalyst. The nature of the active species, also known as active sites, has a direct impact on the surface activity (known as turnover frequency) and the selectivity of the catalyst. This nature is determined by the composition (i.e., type of metal and promoters or bimetallic), size, and shape of the nanoparticles. Reduction of the size of the nanoparticles increases the amount of surface sites per unit weight of metal and generally results in more active catalysts.

The selection of two or more phases to prepare the catalysts will depend on the catalytic process that has to be carried out. For example, for a hydrogenation reaction, the active metal should adsorb and dissociate the molecule of $H_2$ into atomic hydrogen. The atomic hydrogen can hydrogenate a double-bonded hydrocarbon molecule. For a cracking reaction, a high molecular weight hydrocarbon is converted into more valuable compounds of lower molecular weight. This reaction requires an acid solid, such as zeolite loaded with transition metals, for example.

For the $CO_2$ methanation processes various noble (Ru, Rh) and non-noble (Ni, Co, Cu) metals have been extensively investigated as active sites. From the data in the literature, the activity of the metals can be presented as: Ru > Rh > Ni > Fe > Co > Os > Pt > Ir > Mo > Pd [13]. It is also indicated in the case of Ru the high $CH_4$ selectivity and high resistance to oxidizing atmospheres [30]. The main drawback is the cost. Fe is also interesting as a catalyst, mainly due to the reduced cost of the Fe salts. Ni is a metal that combines the advantages of almost all the active metallic phases, with the only drawback being its tendency to oxidize with small amounts of oxygen. The interaction between the support and the metallic active center is also relevant to consider when it comes to catalytic behavior. Many times, metallic oxides are obtained by solid-state reactions that cause the catalytic composting to be modified. Hence, the importance of the characterization of the catalyst and the use of the techniques are described below.

### 2.3. Preparation of the Catalyst

After the selection of the support and metal oxide specific to the catalysis process, catalyst preparation then proceeds using one of the many well-known preparation methods. Among the most used methods for industrial applications are: precipitation methods, impregnation methods, sol–gel methods, and chemical deposition methods. There are other methods for preparing catalysts, all of which involve the idea of efficiently dispersing the active phase on the surface of a material that acts as an inert support.

(a)  Precipitation method: This method is sometimes known as co-precipitation and is one of the most widely used catalyst preparation methods. This method can be used to prepare a single component catalyst or a supported mixed oxide catalyst. The method is based on the precipitation of a single or multi-phase solid by altering the slurry condition, for example, the pH of the solution, applying heat or vaporizing a suitable amount of a precursor, and adsorbing it on a support material. The co-precipitation method slightly differs from the previous one as, in this case, the catalyst is formed by dissolving and mixing the active metal salt and the support to promote nucleation and growth of a combined solid precursor containing both the active element and support.

(b)  Impregnation method: In this method, the selected support is immersed in a precursor solution allowing the precursor of the active phase to diffuse into the porous structure of the support. The obtained slurry is slowly dried and later calcined at an appropriate temperature without exceeding the thermal decomposition temperature of the support to prevent its collapse and sinter its texture.

(c)  Sol–gel method: This method is very versatile and allows control of the texture, composition, homogeneity, and structural properties of solids, and makes possible production of tailored materials such as dispersed metals, oxidic catalysts, and chemically modified supports. This method involves the formation of a sol from dispersed colloid solutions or from some inorganic precursors as a starting material followed by the formation of a gel. This method yields various configurations such as monoliths, coatings, foams, and fibers without using highly cost processing technologies. The method is based on the hydrolysis and condensation of metal alkoxides such as $SiCl_4$ with alcohol [36–38].

(d)  Chemical deposition: This method consists of the formation of thin films on a heated substrate via a chemical reaction of gas-phase precursors, which is known as the Chemical Vapor Deposition (CVD) method. A typical example of a catalyst prepared by this method is a 2D transition metal dichalcogenides deposited on thin polymeric films [39,40].

### 2.4. Characterization Techniques

Some important characterization techniques widely cited in the literature are listed below. For instance: spectroscopic techniques, adsorption techniques, thermal techniques, etc.

#### 2.4.1. X-ray Diffraction (XRD)

This technique gives information about the crystalline phases present in a sample as well as about the bulk properties of the sample (see Figure 1). The X-ray diffraction pattern can also measure the distance between single planes of atoms in a crystal and also provides a measurement of the layer's height. Due to the difference between cell parameters, symmetry, and space groups that crystalline materials have, characteristic diffraction patterns that work as a fingerprint of the material are produced. To understand what conditions are required for diffraction, Bragg's law (see Equation (1)) provides a simplistic model that also is used to calculate the d-spacing. For parallel planes of atoms (with a space $d_{hkl}$ in between) constructive interference can only occur when this law is satisfied:

$$n\lambda = 2 \cdot d_{hkl} \cdot \sin\theta \tag{1}$$

where *d* is the interlayer distance between two consecutive layers, *n* is the diffraction order, $\theta$ is Bragg's angle (formed both by the incident X-ray beam and the diffracted by the planes) and $\lambda$ is the wavelength of the X-rays (0.15418 nm for the copper anode). Additionally, the plane normal (*hkl*) must be parallel to the diffraction vector s (bisects the angle between the incident and the diffracted beam). The crystallite size can also be calculated from the XRD data. It is measured with the Scherrer formulae (see Equation (2)):

$$\tau = \frac{K \cdot \lambda}{\beta_\tau \cdot \cos \vartheta} \tag{2}$$

where *K* is the shape factor = 0.9; $\beta_\tau$ is the width of the peak at half of the maximum intensity (Full Width at Half Maximum, FWHM) after subtraction of instrumental broadening and $\theta$ is the diffraction angle. Normally, the crystallite size is measured in nm which are the units of the wavelength of the X-rays. It must also be taken into account that it is a crystallographic measurement and that it is determined in volume. A particle size can also be determined by other techniques but the material may not need to be crystalline and may be a surface dimension, such as obtained by chemisorption measurements.

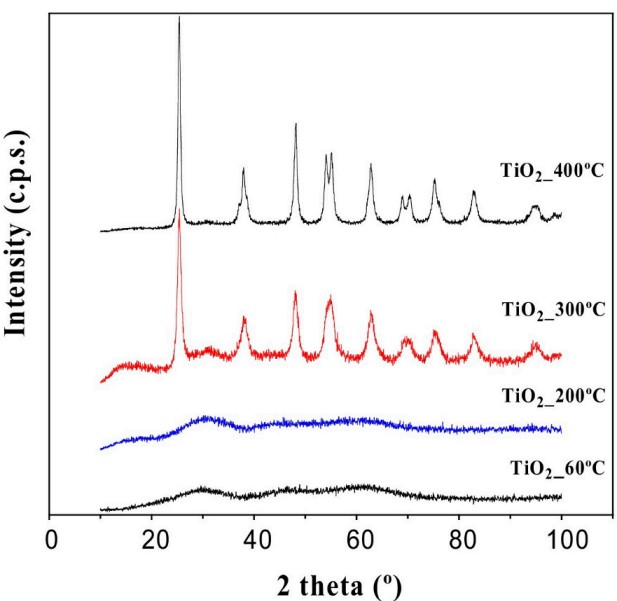

**Figure 1.** XRD patterns of TiO$_2$ treated at several temperatures (reprinted with permission from [41]).

In this figure, it can be seen how the treatment temperature allows the crystalline structure of anatase (TiO$_2$) to develop. At low temperatures, no diffraction peaks are shown because it is an amorphous solid, while the structure is ordered with temperature and allows the characteristic diffraction lines of TiO$_2$ to appear.

2.4.2. X-ray Photoelectron Spectroscopy (XPS) or Electron Spectroscopy for Chemical Analysis (ESCA)

These surface analysis techniques provide valuable quantitative and chemical state information about the surface of the material and its accessibility for reaction [42]. The techniques yield information at an average depth of 5 nm on the surface of the catalyst. The XPS technique is based on exciting a sample surface with monoenergetic Al k$\alpha$ X-ray causing photoelectrons to be emitted from the solid surface. The emitted electron is then collected at an electron energy analyzer. The binding energy and intensity of the photoelectron peak can be used to determine the elemental identity, chemical state, and quantity of the detected element. An example of the XPS spectrum is presented in Figure 2. The XPS spectrum corresponds to an organic waste (ostrich bones) chemically modified by the reduction of an iron salt with NaBH$_4$ to obtain an adsorbent material [43]. The signals

identified in the spectrum provide information about the surface chemical composition. In this case, the presence of oxides of Ca, P, and Fe are observed. In a similar step, in AES the electron emission is used to obtain information about the chemical state of an element. This technique is more sensitive than XPS for lighter elements.

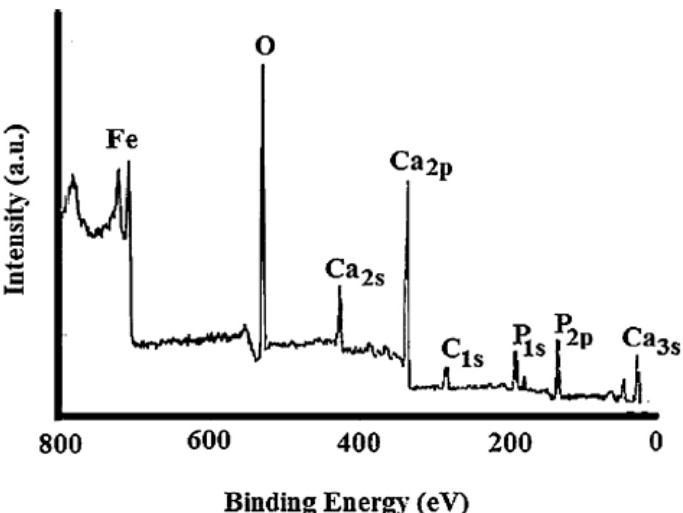

**Figure 2.** Typical spectrum of X-ray photoelectron spectroscopy (XPS) (reprinted with permission from [43]).

### 2.4.3. Ion Scattering Spectroscopy (ISS)

This is another surface technique in which a beam of ions is scattered from the surface of the atom and is gathered by a detector, which determines their kinetic energy [44,45]. Peaks are observed at different kinetic energies and related to the mass difference between the ion and the atom. Thus, two atoms on the surface with different masses will scatter ions differently. Because ions are scattered from the surface, the ISS technique is extremely sensitive and therefore analyzes only the topmost atomic layer on the solid. The ISS technique determines the relative coverage of a surface containing a certain atom or element. Adsorption of certain molecules onto a single crystal follows the growth of ultra-thin layers such as in atomic layer deposition. The downside of the technique is that it requires an ultra-high vacuum and is sensitive to contamination. A typical example of this kind of analysis is summarized in Figure 3.

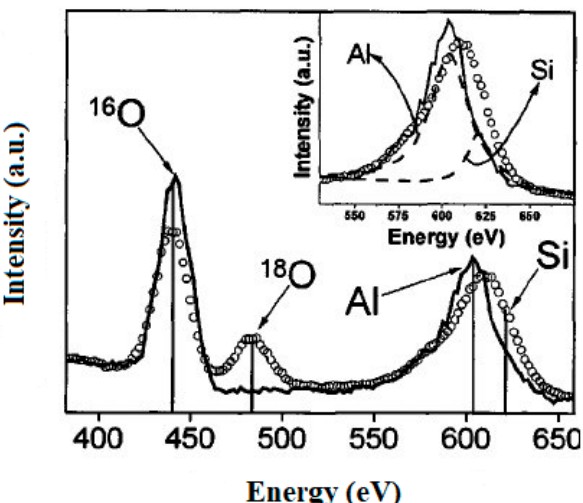

**Figure 3.** Typical spectrum of ion scattering spectroscopy (ISS) (reprinted with permission from [46]).

2.4.4. Electron Microscopy (EM)

Electron microscopes use signals arising from the interaction of an electron beam with the sample to obtain information about structure, morphology, and composition. Electrons are small particles, like photons in light, that act as waves. A beam of electrons passes through the specimen, and then through a series of lenses that magnify the image. The image results from a scattering of electrons by atoms in the specimen. A heavy atom is more effective in scattering than one of low atomic numbers, and the presence of heavy atoms will increase the image contrast.

Scanning Electron Microscopy (SEM) is a technique that uses an electron beam to produce an image of the sample surface with a resolution down to the nanometer scale. Electrons are emitted from a filament and collimated into a beam in the electron source [47]. The beam is then focused on the sample surface by a set of lenses in the electron column. When high-energy electrons reach the sample, several electrons and X-ray signals are generated. An example is included in Figure 4. Using this technique it is possible to morphologically characterize materials and, in this case, it is possible to observe how a ceramic material based on $TiO_2$ is deposited on the surface of a metal wire.

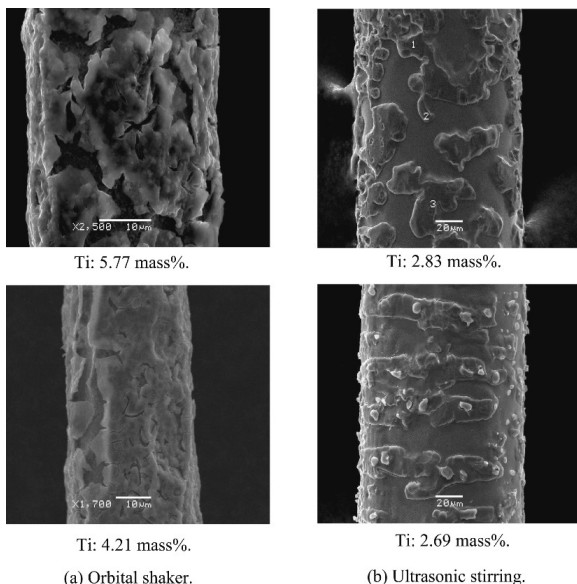

Ti: 5.77 mass%.  Ti: 2.83 mass%.

Ti: 4.21 mass%.  Ti: 2.69 mass%.

(a) Orbital shaker.  (b) Ultrasonic stirring.

**Figure 4.** SEM images of $TiO_2$-coated wires with various stirring methods. The Ti content obtained from EDX analysis is also presented (reprinted with permission from [48]).

These include:

(a) Backscattered electrons (BSE): These are high-energy electrons that are ejected from the solid, losing only a small amount of energy. They originate from deep layers on the surface (a few microns deep). They provide information about the composition of the surface and lower-resolution images. These electrons are reflected after elastic interactions between the beam and the sample.

(b) Secondary Electrons (SE): These electrons originate from a few nanometers into the sample surface, with a lower energy compared to the backscattered electrons. They are very sensitive to surface structure and provide topographic information. Secondary electrons are a result of inelastic interactions between the electron beam and the sample.

(c) X-rays: These characteristic X-rays are produced when electrons hit the sample surface. They give information about the elemental composition of the sample.

BSE comes from deeper regions of the sample, while SE originates from surface regions. Therefore, BSE and SE carry different types of information. BSE images show high

sensitivity to differences in atomic number: the higher the atomic number, the brighter the material appears in the image. SE imaging can provide more detailed surface information.

Transmission Electron Microscopy (TEM) is a very powerful tool for material science. A high-energy beam of electrons is shone through a very thin sample, and the interactions between the electrons and the atoms can be used to observe features such as the crystal structure and features in the structure like dislocations and grain boundaries. Chemical analysis can also be performed. TEM can be used to study the growth of layers, their composition, and defects in semiconductors. High resolution can be used to analyze the quality, shape, size, and density of quantum wells, wires, and dots. A typical TEM analysis is included in Figure 5. In this case, these are images of montmorillonites where it is possible to observe the lamellar structure.

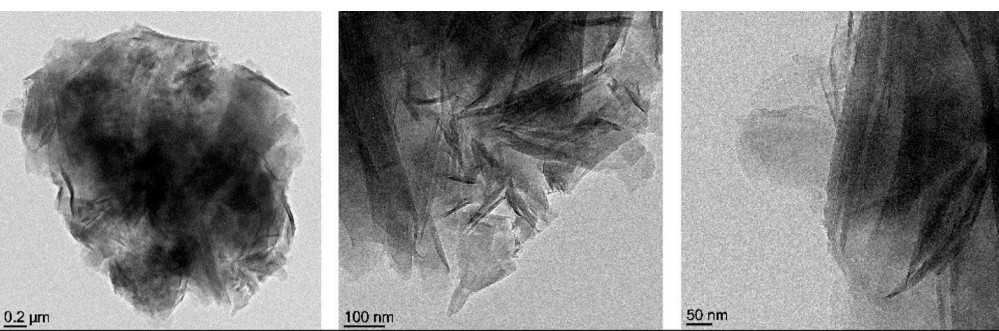

**Figure 5.** Clay image from transmission electron microscopy (reprinted with permission from [49]).

The TEM operates on the same basic principles as a light microscope but uses electrons instead of light. Because the wavelength of electrons is much smaller than that of light, the optimal resolution attainable for TEM images is many orders of magnitude better than that from a light microscope. Thus, TEMs can reveal the finest details of internal structures in some cases as small as individual atoms.

### 2.4.5. Infrared Spectroscopy (IR)

IR spectroscopy works on the principle that molecules absorb specific frequencies that are characteristic of their structure [50]. At temperatures above absolute zero, all the atoms in molecules are in continuous vibration with respect to each other. The IR spectrum of a sample is recorded by passing a beam of IR radiation through the sample. When the frequency of a specific vibration is equal to the frequency of the IR radiation directed at the molecule, the molecule absorbs the radiation. The examination of the transmitted light reveals how much energy was absorbed at each frequency (or wavelength) and this is related to the nature of the element. There are basically two types of spectrometers used in IR spectroscopy–Dispersive IR (DIR) spectrometers and Fourier transform IR (FTIR) spectrometers. Dispersive IR consists of radiation from a broadband source passing through the sample and is dispersed by a monochromator into component frequencies. Then, the beams fall on the detector which generates an electrical signal that results in a recorder response. The Fourier transform infrared (FTIR) technique has replaced the dispersive technique for most applications due to its superior speed and sensitivity. Instead of viewing each component frequency sequentially, as in a dispersive IR spectrometer, all frequencies are examined simultaneously in FTIR spectroscopy. A typical IR spectrum, in this case, the spectrum corresponding to activated carbons, is shown in Figure 6. In this case, it is an active carbon whose surface has been activated by treatment with acids. This treatment makes it possible to generate active centers for the adsorption of molecules as well as for catalytic processes, in principle interesting to stabilize the metallic active phase. Using this technique, functional groups generated on the surface can be detected.

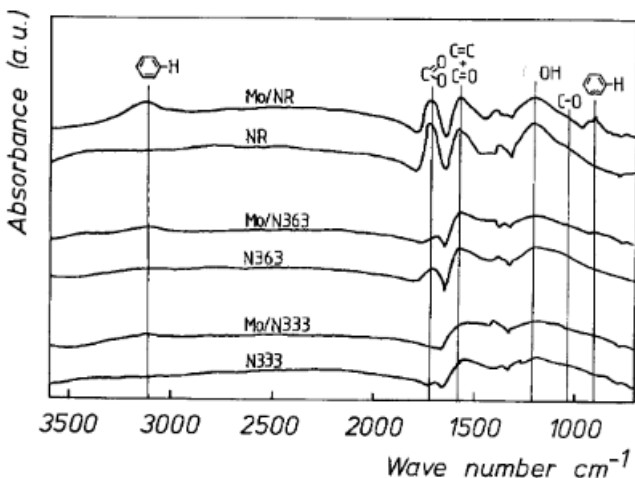

**Figure 6.** Typical IR spectrum of activated carbons (reprinted with permission from [51]).

2.4.6. Mössbauer Spectroscopy

This technique was discovered by Rudolf Mössbauer in 1958 and is based on the absorption of gamma rays in solids due to the vibrations of the atoms [52]. The unique feature of this technique is that absorption occurs only at the nuclei level and has high-resolution energy sufficient to resolve the hyperfine structures of the nuclear level. It operates on a single gamma ray transition, that is the Mössbauer transition, between the ground state and the exited state of one isotope in the sample. This technique is very sensitive to small changes in the chemical environment of certain nuclei. The number, positions, and intensities of the produced peaks in absorbance yield the characterization of the sample. A typical $^{57}$Fe Mössbauer spectrum of clay (Na-Ballarat saponite) is presented in Figure 7.

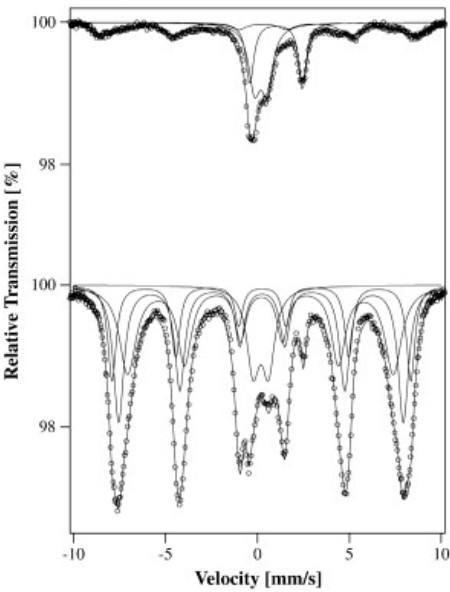

**Figure 7.** Typical $^{57}$Fe Mössbauer spectra at $-268.9$ °C of Na-Ballarat saponite (**top**) and this clay intercalated with Fe8-tacn polycations (**bottom**) (reprinted with permission from [53]).

2.4.7. Thermal Analysis

(a) Thermogravimetric Analysis (TGA): TGA is a method in which a sample's mass is followed over time as the temperature is increased to yield information about physical and chemical properties of a solid such as transition, absorption–adsorption–desorption, etc. It also gives information about thermal decomposition and solid–gas reactions, for example,

oxidation or reduction [54,55]. A typical system consists of a sensitive and precise balance with a sample holder located inside a programmable furnace. As temperature increases, the system monitors the change in mass loss to incur a thermal reaction. This could happen under different atmospheres: ambient, inert gas, oxidation/reducing gases, corrosive gases, vapors or liquids, as well as pressure or vacuum. The result is a collection of mass change versus temperature or time. A typical TGA curve, while the first derivative of this curve determines inflection points useful for further interpretations such as Differential Thermal Analysis known as DTA is included in Figure 8. The weight losses shown in the figure characterize the thermal processes that take place in lamellar double hydroxides (LDH-hydrotalcites). Thus, the first loss in weight at low temperatures can be related to the loss of physisorbed $H_2O$ and the following losses are related to the dehydroxylation of the surface functional groups. Finally, signals appear due to the formation of metal oxides.

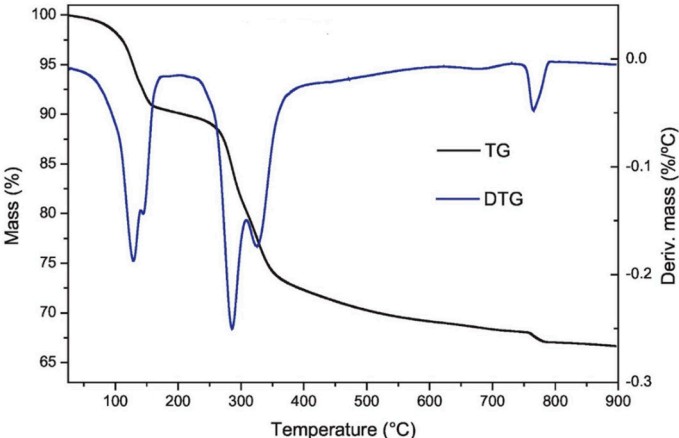

**Figure 8.** Thermal decomposition of hydrocalumite by TGA (reprinted with permission from [56]).

(b) Differential Scanning Calorimeter (DSC): DSC determines the difference in heat flow between the analysis sample and a reference [57]. The results of this technique measure the amount of heat adsorbed–desorbed during a phase transition and the enthalpies associated phenomena. Therefore, it yields information about melting, crystallization, or even water loss from a hydrated sample as well as some characteristic physical and chemical properties of the solid. A typical analysis of DSC is presented in Figure 9.

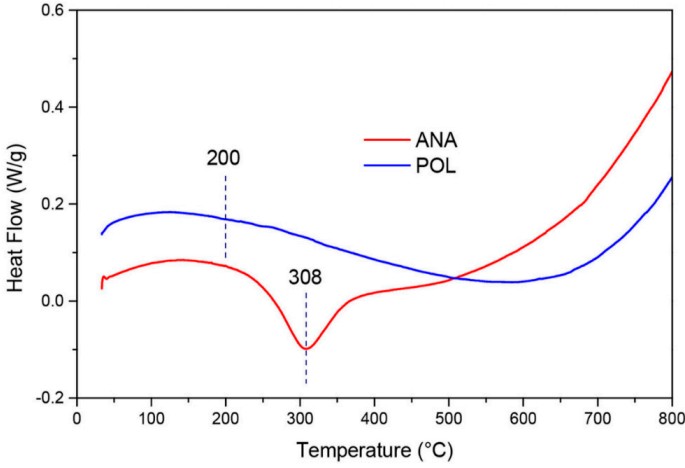

**Figure 9.** Typical profile of DSC on two zeolites (reprinted with permission from [58]).

2.4.8. Adsorption

Adsorption is the selective transfer of certain components of a fluid phase, called solutes to the surface of an insoluble solid. The adsorbed solute is referred to as an

adsorbate, while the solid material is referred to as an adsorbent. When an adsorbent is exposed to a fluid phase, molecules in the fluid phase diffuse to its surface (including its pores if it is a porous adsorbent), where they either chemically bond with the solid surface or are held there physically by weak van der Waals intermolecular forces. When adsorption is caused by van der Waals forces, it is referred to as physical adsorption or physisorption, whereas it is called chemical adsorption or chemisorption if a chemical bond is formed between the adsorbate and the adsorbent. In general, adsorption is a process that involves the accumulation of a substance in molecular species in higher concentrations on the surface of a solid. Adsorption is by nature a surface phenomenon, governed by the unique properties of bulk materials that exist only at the surface due to bonding deficiencies [59–66].

Physical Adsorption Technique: This type of adsorption is due to weak forces between adsorbate and adsorbent. In general, it consists of a solid exposed in a closed space (usually called a sample holder) to a gas or vapors at some definite pressure, and the solid begins to adsorb as soon as the sample holder is immersed in a vessel containing the equivalent liquid of the adsorbate (i.e., $N_2$ gas at liquid nitrogen, Argon at liquid Argon, etc.). Thus, the pressure starts decreasing to reach equilibrium and becomes constant at a value say $P$ where the sample does not adsorb any more gas. The amount of adsorbed gas can be calculated from the decrease in pressure by application of the gas laws providing a known vessel volume. The adsorption is caused by the forces acting between the solid and the molecules of gas. These forces are the same nature as the van der Wals order (<80 kJ/mol) and are also known as Intrinsic Surface Energy which brings about condensation of the fluid gas to the liquid state. The amount of gas absorbed by the sample ($n$) is proportional to the mass of the sample and depends also on the temperature of the bath where the sample is immersed, the pressure of the adsorbate over the sample, and the nature of both the solid and the gas. The quantity of gas adsorbed can be expressed as $n = f(P, T, gas, solid)$, and if the temperature is below the critical temperature of the gas, the above form is rather written as $n = f(p/p°)_T$, *gas, solid*, with $p°$ being the saturation vapor pressure of the adsorptive. A plot of $n(\text{cm}^3/\text{g})$ versus $p/p°$ at a constant temperature yields the adsorption isotherm. The shape of an adsorption isotherm is directly related to the texture of the solid in question. There are six different types of adsorption isotherm, according to BDDT (Brunauer, Deming, Deming y Teller) as shown in Figure 10. Through this classification, it is possible to know if the materials are microporous, mesoporous or microporous solids.

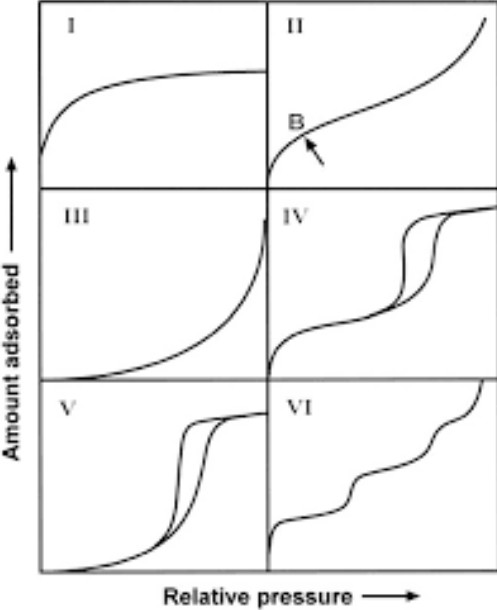

**Figure 10.** Adsorption isotherms according to BDDT classification [61].

The following important properties can be derived directly from the adsorption–desorption isotherm:

(a) Hysteresis: Hysteresis is defined as the void space between the adsorption and the desorption branch of the isotherms corresponding to the same solid and is associated with the capillary condensation that occurs when the relative pressure over the sample exceeds 0.4, approximately. The hysteresis is related to the shape of a pore in the solid and occurs when a greater pressure change is required to remove an adsorbate from a pore than what was required to adsorb.

(b) The specific surface area: Langmuir assumption [67,68] applies for type I isotherm and corresponds to microporous solids (<2 nm in pore diameter) and BET [69] applies for type II and IV (non-porous and mesoporous, respectively, pore diameter $2 < d < 50$ nm).

(c) *t-plot* analysis: According to Lippens and de Boer [70,71], a comparison of the adsorption isotherm produced by a solid with a standard isotherm (produced by a non-porous material but with a similar nature of the solid in question) yields information about the micropore volume as well as the external surface area of the solid. The external surface area corresponds to the area in the solid that extends above the micropore area. According to Halsey, one of the t-equations, the t value is evaluated from the inverse of the relative pressure according to Equation (3).

$$t = 3.53 \left[ \frac{5}{\ln(p°/p)} \right]^{1/3} \tag{3}$$

(d) Pore size and structure: This term is related to mesoporous materials and is directly connected with type IV of adsorption isotherm [71], which shows a hysteresis between the adsorption and the desorption branch. According to Kelvin, there is a direct relationship between the size of the pore and the relative pressure in equilibrium withdrawn directly from the adsorption isotherm (Equation (4)). Finally, the pore size would be the sum of the Kelvin radius and the value of the thickness (*t*) according to the Pierce method (see Equation (5)) [72–74].

$$r_K = \frac{-2 \cdot V_m \cdot \gamma \cdot \cos\varphi}{RT \cdot \ln(p/p°)} \tag{4}$$

$$r_p = r_K + t \tag{5}$$

where $r_K$ represents the radius of the pore, $\gamma$ is the surface tension of the adsorbate, $V_m$ is the molar volume, $R$ is the gas constant, and $T$ the analysis temperature.

(e) Total pore volume: The total pore volume of a solid corresponds to the total adsorbed volume at a relative pressure close to unity on the adsorption isotherm, it is normally converted into volume of liquid according to the Gurvitsch rule (see Equation (6)) [75].

$$V_{pT} = V(g) \cdot 0.0015468 \tag{6}$$

Chemical Adsorption Technique: Chemisorption is caused by a reaction on an exposed surface, which creates an electronic bond between the surface and the adsorbate. The heat of chemisorption can exceed 1000 kJ/mole and shows a type I adsorption isotherm. Chemisorption, therefore, consists of one layer of adsorbate on the surface where active sites are located. Hence, the total adsorbed amount of adsorbate determined at saturation corresponds to the number of active sites available and accessible on the surface of the solid providing a known stoichiometry of adsorption between the adsorbate molecule and the active site. Chemisorption encompasses the following important techniques.

Pulse chemisorption technique. This technique serves to titrate the surface-active sites available for the reaction. It is based on dosing a known amount of active gas, normally $H_2$ or CO, onto a freshly reduced catalyst. The total available surface species are thus quantified providing the stoichiometry factor (SF) of adsorption. SF is normally considered to be 1

for CO and 2 for $H_2$. Pulse chemisorption evaluates the accessible surface of materials and can quantify the dispersion of the active species on the support. The higher the dispersion the higher the activity of the catalyst. The pulse technique, therefore, provides a rapid comparison of the catalysts and provides a prediction that can be extrapolated to the overall performance of the catalysts. This technique is also widely used to evaluate the total acidity of the catalyst when ammonia or any other basic molecules are used for titration.

2.4.9. Temperature Programmed Reduction (TPR)

This important technique consists of reducing a metal oxide under the effect of a reducible gas, $H_2$ or CO, and temperature to produce a result known as the TPR profile. Important information can be withdrawn from the TPR profiles and can be enumerated as follows:

(a)   The quantity of the total amount of oxides in the solid;
(b)   The maximum reduction temperature of the oxides;
(c)   The presence of various particle sizes of the oxide that would be indicated by shoulders on the TPR profile;
(d)   Shift on the maximum reduction peak indicating the interaction of the active particles with the support. In other words, the role of the support is to stabilize the active species by a certain type of interaction that could be weak or strong; the stronger that interaction is the higher the reduction temperature.

Percentage of reducibility: This indicates the number of oxides that are free to be reduced and become active sites on the solid. The difference between the originally loaded material at preparation and the number of reducible species calculated by TPR indicates the number of species that are capable of being reduced.

Identification of species: TPR is a technique that allows the identification of new species where two or more elements are present in the catalyst. A typical example of when two species are interacting or reacting to produce a completely different species is included in Figure 11. In this case, $PtO_2$ and $RuO_2$ are mixed physically and the TPR profiles show a mixture of two profiles that correspond to each individual element. If these two elements are subject to a high temperature, they react to form an alloy. Therefore, the TPR technique demonstrates how different species react to produce new species, in this case, an alloy, and therefore its presence could also be identified in a catalyst formed by these two metal oxides.

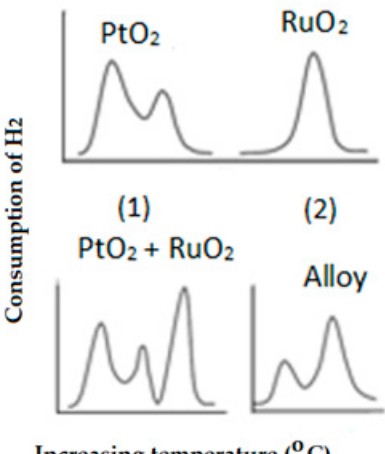

**Figure 11.** TPR profiles of individual $PtO_2$ (1) and $RuO_2$ (2) and the physical mixture ($PtO_2$ + $RuO_2$) after thermal treatment to obtain the metal alloy.

2.4.10. Temperature Programmed Oxidation (TPO)

This technique is used to oxidize the freshly reduced elements in a catalyst. As TPR is a bulk reduction technique, which means that all oxide species present in the catalyst

area are completely reduced under the effect of a reducible gas and high temperature. TPO evaluates the freshly reduced species to being re-oxidized. Thus, determining the degree of reduction. Cycles of reduction/oxidation, however, can predict the active lifetime of a catalyst after several regeneration cycles. This analysis would produce CO or $CO_2$ as the result of carbon or graphite oxidation. A mass spectrum of CO and $CO_2$ is included in Figure 12. Carbon deposits on the active sites of the catalyst lead to deactivation which directly results in lowered yield.

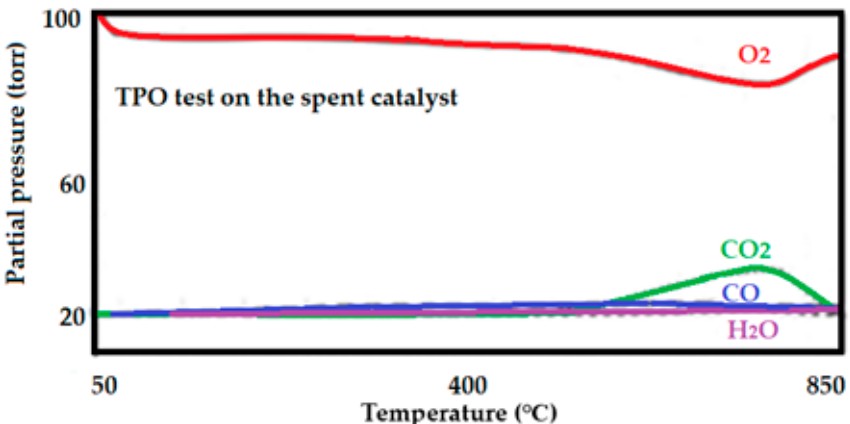

**Figure 12.** Mass signal of the consumption of $O_2$ and those corresponding to the production of CO and $CO_2$.

### 2.4.11. Temperature Programmed Reaction (TPRe)

This technique shows a simple reaction performed at atmospheric pressure. It helps to simulate and predict a reaction that would produce a larger yield at industrial levels. The following figure shows an interesting reaction in which $CO_2$ is sequestrated by CaO to produce a more beneficial product such as Calcium carbonate $CaCO_3$. The result shown in Figure 13 indicates the number of cycles that CaO can react with $CO_2$ before being deactivated. After 4 cycles, the area that was produced on the first cycle is about 4 times the area produced on the 4th cycle. Such that, the number of cycles is related to the capacity of the solid, in this case, the CaO, to react and sequestrate the $CO_2$. This result provides insight into the reusability of a catalyst.

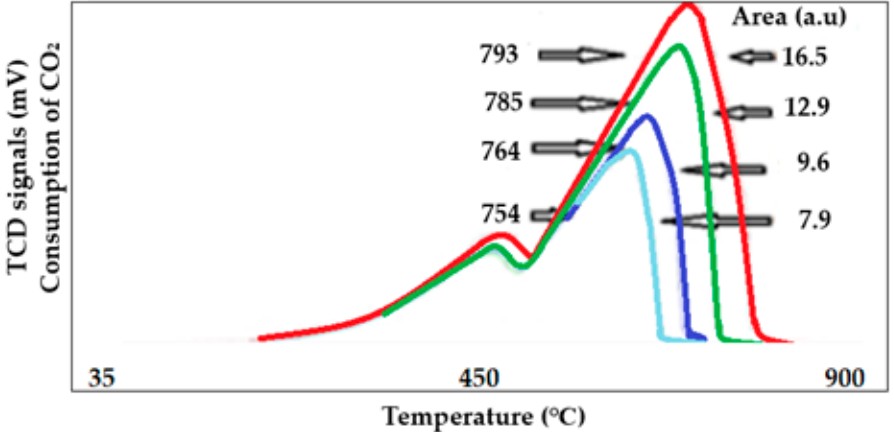

**Figure 13.** Sequestration cycles of $CO_2$ by CaO.

### 2.4.12. Evaluation of Acid Sites in Solids

Some catalytic reactions require acid sites instead of metal active sites. One example is in the cracking reaction. The strength and distribution of acid sites are rather important

in catalysis. In this experiment, the sample is saturated with a basic molecule such as ammonia, isopropylamine or any other branched basic molecule. The experiment is carried out at about 50 to 70 °C to minimize the physisorption effect that can occur on the support. Distribution and strength of acid sites, normally Brønsted acid sites (BAS), is then performed by raising the sample temperature which causes desorption of the pre-adsorbed reactants as a function of temperature. If desorption is performed at several temperature ramping rates (2, 5, 10, 15, and 20 °C/min), the heat of desorption can then be determined by applying the Kissinger Equation (7) [76] that is related to the strength of acid sites. An example of the acidity of ZSM5 zeolite evaluated after saturation by ammonia followed by desorption at various ramping rates is presented in Figure 14. This analysis yields very important information about the performance of a catalyst used for cracking. In catalytic cracking, the acid sites are the active sites that break down large hydrocarbon chains into aromatic molecules that are of great interest in the fuel industry. Evaluating the desorption capacity at various heating rates allows obtaining the energy related to the desorption process, that is, the energy necessary for the desorption of the adsorbate.

$$2lnT_m - ln\beta = \frac{E_d}{R \cdot T_m} + C \tag{7}$$

$$C = ln\left(\frac{E_d \cdot V_m}{R \cdot K}\right) \tag{8}$$

where $T_m$ (°C) is the maximum temperature of the temperature-programmed desorption curve, $\beta$ (°C/min) is the heating rate, $E_d$ (kJ/mol) is the heat of desorption and $R$ is the gas constant.

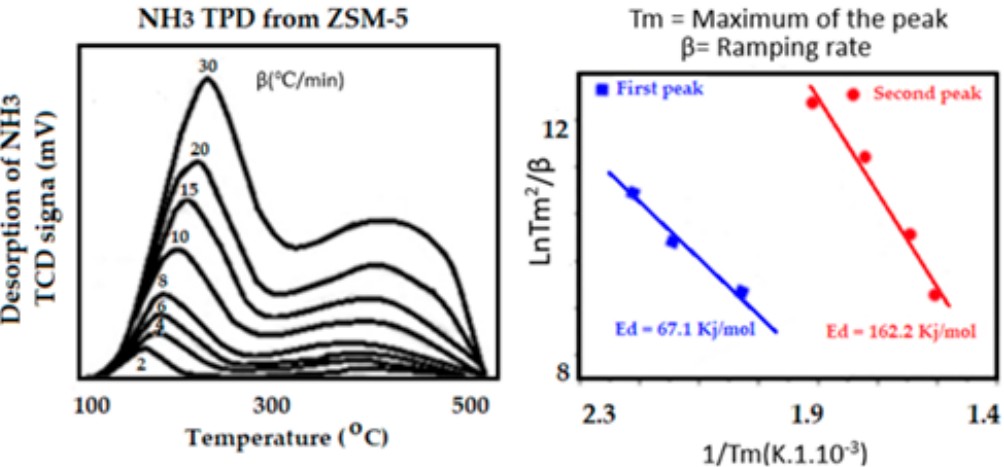

**Figure 14.** Temperature desorption profiles of $NH_3$ are used to determine the heat of desorption and the distribution of acid sites according to their strength.

Ammonia as a small probe molecule can penetrate all size pores, micro and mesopores without discrimination. However, in many cases, the very tiny pores will not make any contribution to the overall activity of the catalyst. They, on the contrary, prevent the adsorbate molecules from diffusing inside such a small pore to encounter the active species and react. In this case, larger and more branched base probe molecules are more suitable. I-propylamine as a probe molecule quantifies acidity only in larger pores that make a contribution to the activity of the catalyst. This probe molecule absorbs, reacts, and decomposes over a BAS to produce propylene and other products. The amount of the produced propylene is equivalent to the amount of BAS in the catalyst. The decomposition of the i-propylamine takes place in a similar manner to the Hofmann elimination [77–79] according to the following schematic for the reaction (see Figure 15).

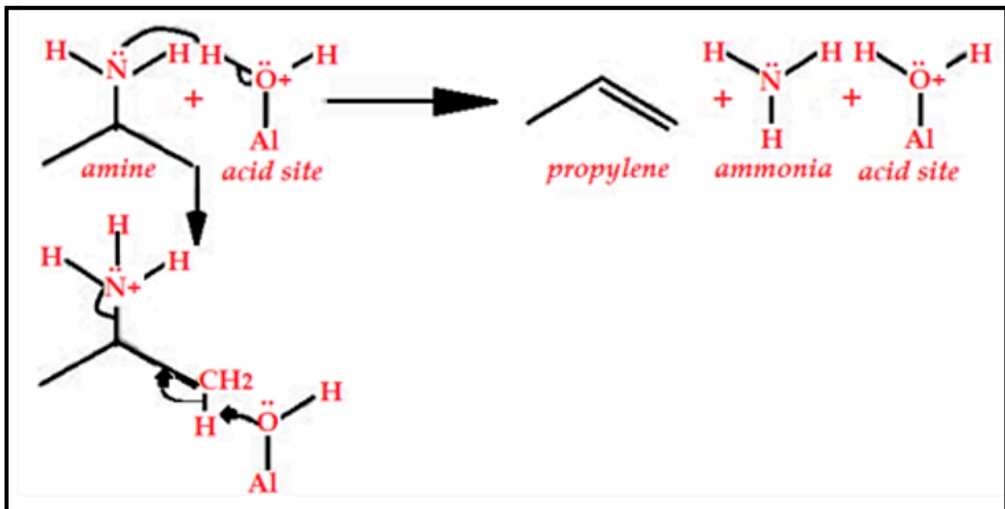

**Figure 15.** Reaction of the amine with the acid site decomposes to produce propylene and ammonia. The mechanism is similar to the Hofmann elimination.

Methanol is also a suitable molecule to determine active sites on the catalysts [80]. It produces various products according to the active site on which decomposition takes place. A possible mechanism and the products of the reaction are included in Figure 16.

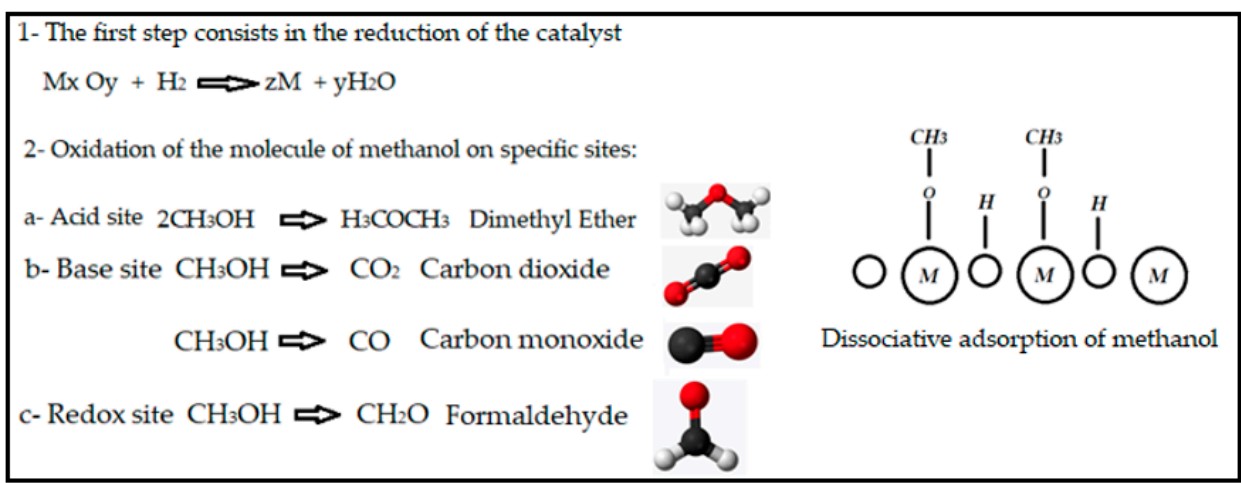

**Figure 16.** Decomposition of the molecule of methanol over an active site.

### 2.4.13. Mercury Intrusion

This technique is used to characterize solids that contain pores (0.003–360 µm), where the gas adsorption technique fails to study such large pores. This technique uses mercury to penetrate under pressure the pore. Mercury is a non-wetting liquid; therefore, the instrument applies pressure to provoke mercury to penetrate the pore. The size of the pore is related to the applied pressure according to the Washburn [81] Equation (9).

$$p = \frac{-4 \cdot \gamma \cdot cos\theta}{d} \qquad (9)$$

where $p$ represents the applied pressure, $\gamma$ is the surface tension of the adsorbate (mercury in this case), $\theta$ is the contact angle and $d$ is the pore dimension.

## 3. Synthesis, Characterization and Catalytic Application

As an example of this work, two series of Ni-supported catalysts were prepared using the impregnation method. The commercial supports used were an alumina trilobe (high surface area material, 310 m$^2$/g) and an alumina pellet (lower surface area material, 92 m$^2$/g). Two series of catalysts were prepared on each of the supports, one was with low-loading metal (about 5 wt.%) and the other series was with a higher loading of Ni (about 14 wt.%). Equivalent amounts of salt (Ni(NO$_3$)$_2$·6 H$_2$O from Aldrich) were taken and dissolved in a volume of DI water corresponding to three times the total pore volume of the support determined from the adsorption isotherm at a relative pressure close to unity. The supports were first degassed at 150 °C for 3 h to completely dry. The samples were then brought to room temperature under an inert atmosphere. The solution was poured over 5 g of the dried support and stirred for 6 h until the solution was completely evaporated. Later, the fresh sample was washed with DI water until the water came out clear. Samples were then placed in the fixed bed reactor where the catalytic tests were carried out and dried at 100 °C by heating the oven at a rate of 1 °C/min and maintained at 100 °C for 12 h. Calcination was carried out by flowing a mixture of 10% O$_2$ in helium and heating the reactor up to 500 °C with a rate of 2 °C/min and maintained for 12 h.

### 3.1. A Comprehensive Assessment of Characterization Techniques

Both the characterized supports and the freshly prepared catalysts were analyzed by gas adsorption using N$_2$ as the probe molecule at −196 °C (Micromeritics 3flex apparatus, Norcross, GA, USA). The sample was dried at 150 °C under vacuum for 4 h prior to the analysis.

Temperature Programmed Reduction (TPR) (Micromeritics in situ characterization system (ICCS) connected to the Micromeritics FR-100 microreactor): This technique is used to study the role of the support in stabilizing the active species. Ni with various loading on different supports as well as a commercial copper catalyst (13 wt.% Cu-alumina) was used for comparison with the prepared Ni catalysts.

Pulse chemisorption of CO: This technique was used to titrate the surface atoms or active particles present on the surface of the catalyst at a temperature near 35 °C. Carbon monoxide was used as adsorbate and a stoichiometry factor of 1 was considered to correlate the amount of CO chemisorbed to the number of surface-active particles.

N$_2$O was used to determine the dispersion of Cu on the Cu-supported catalyst. The CO was substituted by N$_2$O since Cu does not chemisorb CO or H$_2$. The catalyst was first reduced with H$_2$ by performing a TPR analysis to ensure a complete reduction of the metal. The titration is carried out at 90 °C for a complete surface oxidation of Cu. This analysis produces N$_2$ as the result of Cu oxidation. Quantification of active sites is determined by computing the amount of H$_2$ produced in relation to the stoichiometry factor and the amount of Cu particles on the surface of the catalyst.

Testing the catalysts for the Sabatier reaction was carried out on the Micromeritics FR-100 flowing reactor (see Figure 17). Approximately 0.5 g of catalyst was placed in a fixed bed reactor and reduced with pure H$_2$ at 500 °C for one hour at 30 bar. The reactor temperature was brought to 30 °C prior to flowing the active gases for the reaction. A mixture of 50 cm$^3$/min of CO$_2$ and 200 cm$^3$/min of H$_2$ was premixed before passing through the catalyst bed at room temperature. The reactor temperature was then raised to 500 °C at a rate of 2 °C/min. The mass spectrometer was connected to the exhaust of the reactor and served to online monitor the signals for both the flowing mixture as well as for the expected products. In this case, 4 masses (*m/z*) were selected for this analysis: (2 for H$_2$, 44 for CO$_2$, 28 for CO, and 16 for CH$_4$).

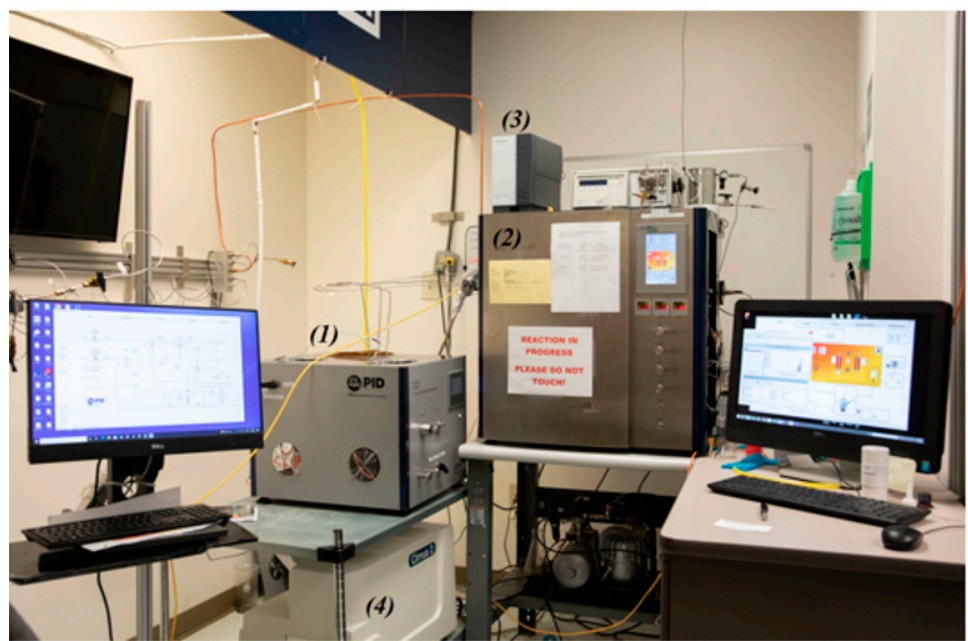

(1) *In-situ* catalyst characterization system (ICCS)
(2) FR-100 Micromeritics micro-reactor
(3) Agilent Micro GC
(4) MKS mass spectrometer

**Figure 17.** Catalytic equipment.

### 3.1.1. Physical Adsorption

The isotherms that correspond to both the supports (red) and to the catalysts as well as for the commercial 13 wt.% CuO-supported alumina catalyst (blue) is presented in Figure 18. Both supports showed type IV adsorption isotherms, indicating that both are mesoporous materials. The surface area as well as total pore volume determined from the adsorption isotherms are summarized in Table 1.

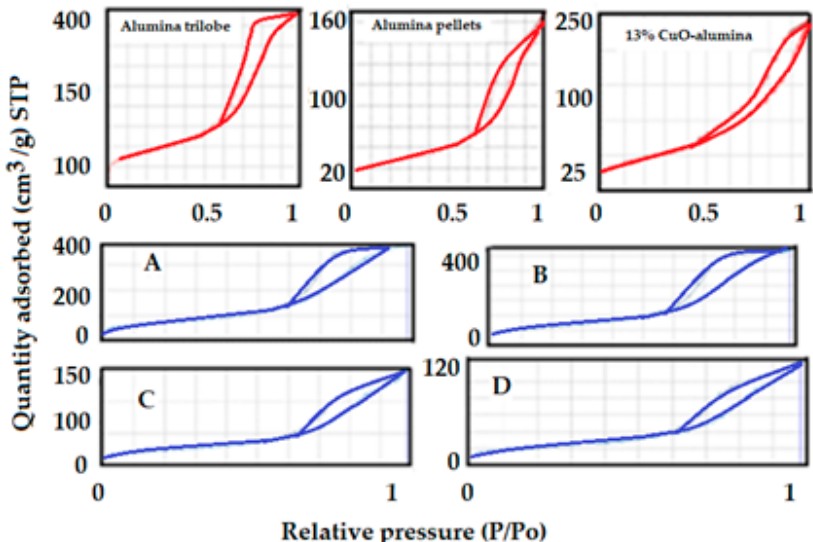

**Figure 18.** $N_2$ adsorption isotherms for all supports and all catalysts as well as for the 13 wt.% CuO-alumina catalyst. (**A,B**) Ni/$Al_2O_3$ trilobe, (**C,D**) Ni/$Al_2O_3$ pellet catalysts.

**Table 1.** BET-specific surface areas, total pore volumes, and pore diameter for all materials.

| Sample | $S_{BET}$ (m²/g) | $V_{pT}$ (cm³(STP)/g) | Pore Diameter (nm) |
|---|---|---|---|
| Alumina Trilobe | 310 | 0.67 | 8.6 |
| Alumina pellet | 92 | 0.26 | 10.9 |
| 13 wt.% CuO-alumina catalyst | 165 | 0.38 | 9.1 |

It can be concluded from the adsorption isotherm that the alumina trilobe contains smaller pores than the other support since condensation occurs at a lower relative pressure (0.6 versus 0.7 $p/p°$). The hysteresis shown in the adsorption isotherms for alumina trilobe reflects ink-bottle pores while the alumina pellets and the copper catalyst showed more open slit-shape pores. The specific surface areas as well as total pore volumes of the supported catalysts showed slightly lower values due to the fact that the Ni penetrated the pores, thus decreasing the pore volume and the internal surface area within the pores. This phenomenon is expected and desirable in catalysis as it stabilizes and increases the dispersion of nickel yielding a higher performance of the catalyst when the active element enters the pore. It should be noted that the surface area within the pore is the internal surface area of the solid and has a large contribution to the total surface area, hence a better dispersion of the active elements inside the pores. Results are shown in Table 2.

**Table 2.** BET specific surface areas, total pore volumes, and pore diameter for all catalysts.

| Catalyst | $S_{BET}$ (m²/g) | $V_{pT}$ (cm³(STP)/g) | Pore Diameter (nm) |
|---|---|---|---|
| 14.5 wt.% Ni (A) | 222 | 0.51 | 8.9 |
| 4% wt.Ni (B) | 280 | 0.53 | 8.7 |
| 11% wt.Ni (C) | 75 | 0.19 | 10.0 |
| 5.6% wt.Ni (D) | 83 | 0.21 | 10.4 |

The surface area was determined by solving the BET equation at relative pressure between 0.05 and 0.3. The pore size was estimated by the Kelvin equation (Equation (4)) and corrected by adding the t value according to de Boer et al. (Equation (3)) to the Kelvin radius. The total pore volume was estimated from the adsorption isotherm at a relative pressure close to 1 and transformed into liquid by using the Gurvitsch rule (Equation (5)).

3.1.2. Chemical Adsorption, Chemisorption or Selective Adsorption Technique

Characterization of the Ni-supported catalysts prepared for this study is used to show examples of the use of some of these techniques.

In general, the low loading in Ni yields a higher dispersion (A: 4 wt.% Ni) of 25.5% while the catalyst (B: 14.5 wt.% Ni) larger loading in Ni yields a much lower dispersion (5%) (see Figure 19A,B). This could happen since large loading could provoke the formation of large agglomeration of Ni, and, hence, lower dispersion. The CO pulse chemisorption profiles for the catalyst supported on alumina pellets are included in Figure 19a,b. As the support in this series of catalysts has a much lower surface area, dispersion also showed much lower values for the high loading (A: 11 wt.% Ni), the dispersion was 3.3%, while the low loading (B: 5.6 wt.% Ni) was 14.5%.

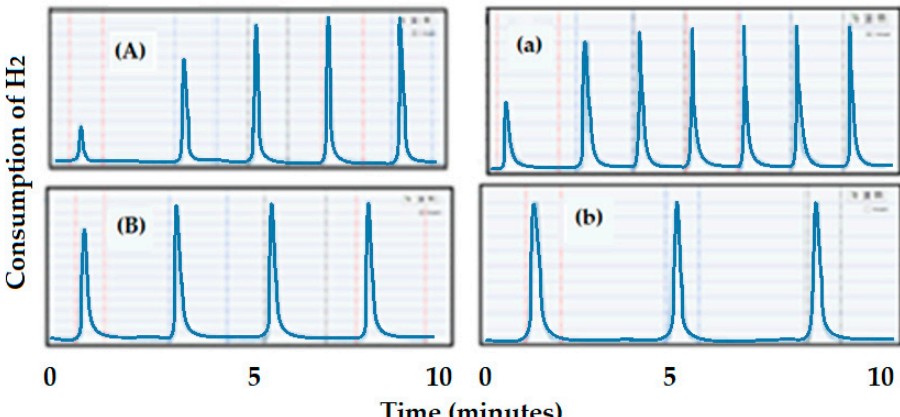

**Figure 19.** CO pulse chemisorption on Ni catalysts. Ni-alumina trilobe catalysts: (**A**) (25.5% dispersion) and (**B**) (5% dispersion). Ni-alumina pellets catalysts: (**a**) (3.3% dispersion) and (**b**) (14.5% dispersion).

The result of the dispersion on the fresh catalysts would predict a higher conversion for the catalysts-supported alumina trilobe. Thus, a higher surface area support and lower loading catalysts produced higher dispersion, and therefore, they would yield a higher conversion of $CO_2$ to $CH_4$.

This analysis was used to characterize the prepared catalysts of Ni supported on alumina, trilobe, and pellet. The role of the support (in this case, supported Ni on high surface area material, alumina trilobe, and other aluminas (pellets) with a lower surface area) and their interaction with the active species is presented in Figure 20. Profile (a) indicates strong interaction between the support and the active species due to the higher reduction temperature, while (b) shows weaker interaction since a lower reduction temperature is shown on the TPR profile. Sometimes the support can strongly interact with the active species, when this occurs, the reaction of the analysis gas on the active species can be disguised up to temperatures of 1000 °C. This strong interaction forms a species called spinels. Spinels are chemically and thermally stable materials and are, generally, not desirable in catalysis except for some biomass reactions. Outside of these biomass reactions, the formed spinels are not as active as the active species and lead to slower product formation.

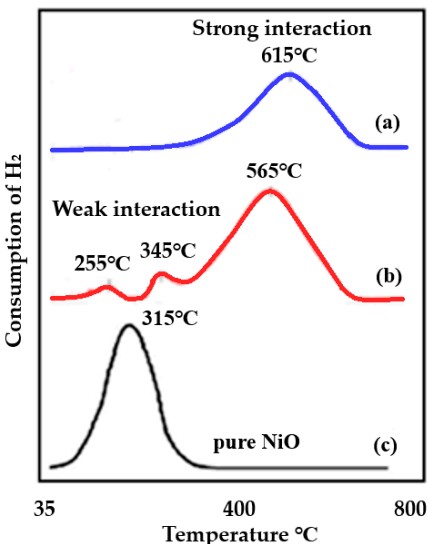

**Figure 20.** TPR profiles showing the role of the support. (**a**) Ni-trilobe, showing strong interaction with the support compared with unsupported NiO, (**b**) Ni-pellets showing weaker interaction with the support compared with unsupported NiO, and (**c**) pure NiO.

Several TPR profiles of Ni-supported alumina catalysts that have been calcined at various temperatures, between 500 and 1100 °C, under a flow of 10% $O_2$ in helium are included in Figure 21. It can be observed for these profiles that the calcination temperature can alter the reduction profiles. A higher calcination temperature tends to form new species, this must be avoided to minimize the formation of spinels and to stabilize the active species under the high temperature and pressure reaction conditions.

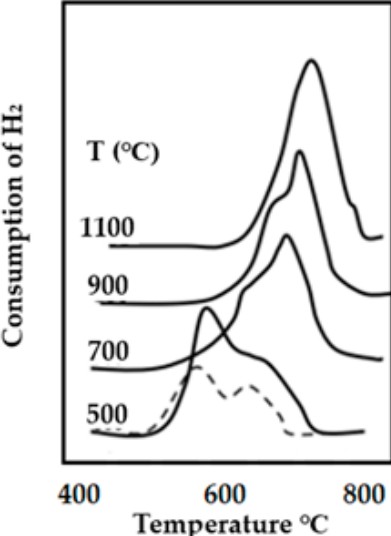

**Figure 21.** TPR profiles showing the effect of the temperature calcination on the shift of the maximum of the reduction temperature. The TPR profile of the dried catalyst precursor is also included (dried line).

Evaluation of the Cu-alumina commercial catalyst by $N_2O$. For the copper catalyst, nitrous oxide was used to determine the dispersion of Cu since the metal would not adsorb/react with $H_2$ nor with CO [82]. In this case, the thermal conductivity detector (TCD) on the ICCS instrument used for the characterization is not capable of separating and identifying the $N_2$ peak from the peak of $N_2O$ that has not been reacted. To overcome this problem, a Propak separation column from Agilent was installed just before the TCD. This column separates the peak of $N_2$ as the product of the reaction (see reaction below) from the peak of $N_2O$ as an excess of the pulse that has not been completely consumed by the copper. Quantification of the amount of $N_2$ produced corresponds to the amount of copper present on the surface of the catalyst providing a stoichiometry factor of 2.

$$N_2O + 2Cu \rightleftarrows Cu - O - Cu + N_2 + (\text{non reactive } N_2O) \tag{10}$$

The catalyst was first reduced under a flow of pure $H_2$ at 350 °C and was kept for 1 h at this final temperature to ensure complete reduction of the copper. After completion of the reduction, the catalyst was swept by a current of helium to completely remove the $H_2$. For the analysis, the sample temperature was reduced to 90 °C, then pulses of $N_2O$ were carried out. The installed column was able to separate the produced $N_2$ from the $N_2O$. The registered pulse of $N_2O$ as well as $N_2$ are summarized in Figure 22. Quantification of $N_2O$ was done upon saturation, where several $N_2O$ peaks produced the same area. The ICCS program monitors the loop temperature and pressure so that the quantification of the loop volume is corrected at each pulse. Having the corrected loop volume at each pulse, the quantification is done by comparing the resultant area peak at each pulse with the peak area at saturation. The freshly reduced copper catalyst produced a dispersion of 29%.

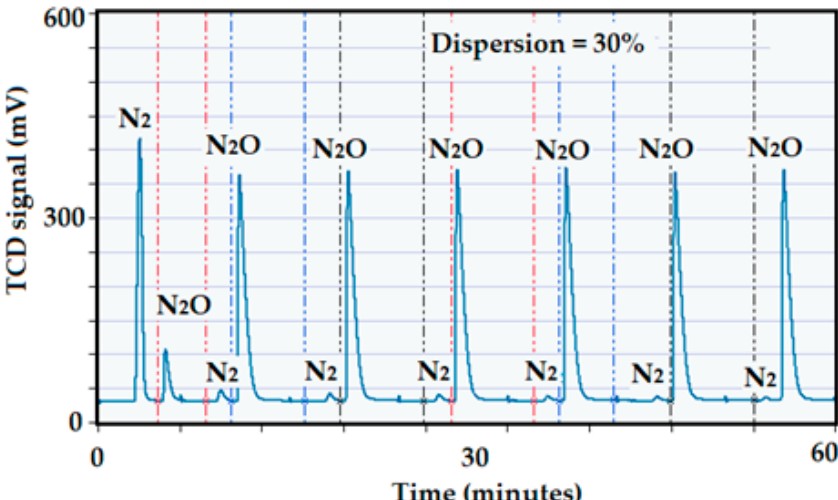

**Figure 22.** Pulse chemisorption of $N_2O$ on the reduced Cu-catalyst.

Upon characterization of the catalysts, they were then tested for the Sabatier reaction, where $CO_2$ was reduced by $H_2$ at elevated temperature and pressure to mainly produce $CH_4$ with some CO as a co-product and other products. The reaction conditions were reduction of $CO_2$ by $H_2$ at 15 bar and heated to 500 °C of temperature with a temperature ramping of 2 °C/minute. Consumption of $CO_2$ and $H_2$ as well as production of CO and $CH_4$ were online recorded by using a mass spectrometer.

Only three catalysts will be shown here as examples: (A) 13 wt.% CuO-alumina, (B) 11 wt.% Ni-alumina pellets and (C) 4 wt.% Ni-alumina trilobe. Before any catalytic test, a blank was performed to ensure that the reactor showed little or no activity for this reaction. The catalyst support was made of quartz wool instead of the normal filter that comes with the reactor, thus, any activity will be attributed to the catalyst.

(A) Catalytic test on the 13 wt.% CuO-alumina catalyst: The reaction conditions were the same as previously described. The analysis was performed using 0.5 g of sample and heating from near room temperature to 550 °C at a rate of 2 °C/min and 15 bar. The produced signals for the products ($CH_4$ and CO) as well as signals corresponding to the consumption of $H_2$ and $CO_2$ are included in Figure 23. The first signal ($m/z$ 28) appeared at about 230 °C and corresponded to the production of CO, while a second signal ($m/z$ 16) appeared at about 360 °C and corresponded to the formation of $CH_4$. The reaction stabilized at about 525 °C, beyond this temperature the analysis stopped to avoid complete deactivation of the catalyst. This deactivation could be due to excess formation of graphite as the co-product of the reaction, thus, minimizing the contact of the reactant molecules with the active area of the catalyst.

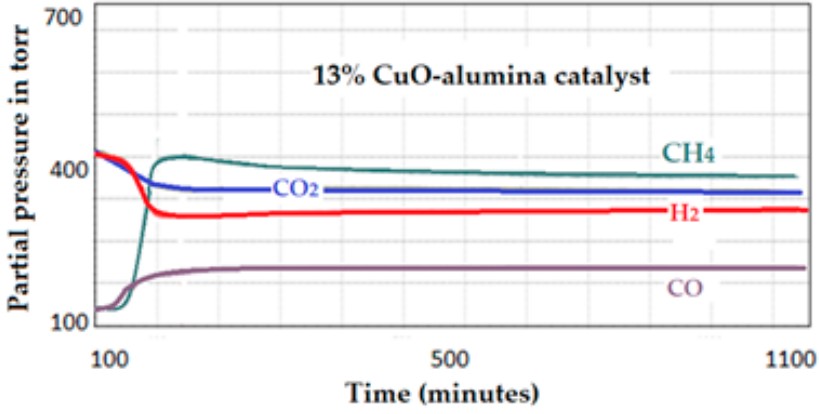

**Figure 23.** Intensities of the produced signals by the 13 wt.% CuO-alumina catalyst.

The performance of all catalysts was estimated as follows:

(1) The intensity difference of $CH_4$ at the end of the reaction minus the intensity at the start (IT);
(2) Determination of the total amount of the metal on the catalyst (MT);
(3) Determination of the amount of the metal on the surface (MS): multiplying the total amount of metal (MT) by the % dispersion of the metal on the catalyst;
(4) Estimation of the performance obtained by dividing the resulting intensity of the expected product (IT)/MS.

Example of the calculation for the 13 wt.% CuO-catalyst:
Activity of the 13% Cu-alumina catalyst:
Initial Intensity of $CH_4$ = 41 Ending intensity = 351
Therefore IT = 351 − 41 = 310
Sample mass = 0.6215 × 13% × 29% = 0.0234
Total amount of Cu in the catalyst: 0.6215 × 13% = 0.0808 g (MT)
Total amount of Cu on the surface: 0.0808 × 29% = 0.0234 g (MS)
Activity of the catalyst to produce $CH_4$ is: 310 ÷ 0.0234 = 13,248

For comparison reasons, the following are several catalysts containing various loads of Ni-supported alumina that were prepared, characterized, and tested the same way as the Cu commercial catalyst.

(B) Catalytic test on the 11 wt.% Ni-alumina pellets. The signals produced by the 11 wt.% Ni-pellets catalyst are summarized in Figure 24. The relative performance was determined as before, by dividing the difference in intensities (end intensity−starting intensity) by the number of surface nickel particles:

400 − 22 = 378
Surface Ni = 0.00247
Relative intensity (activity): 378 ÷ 0.00247 = 153,036

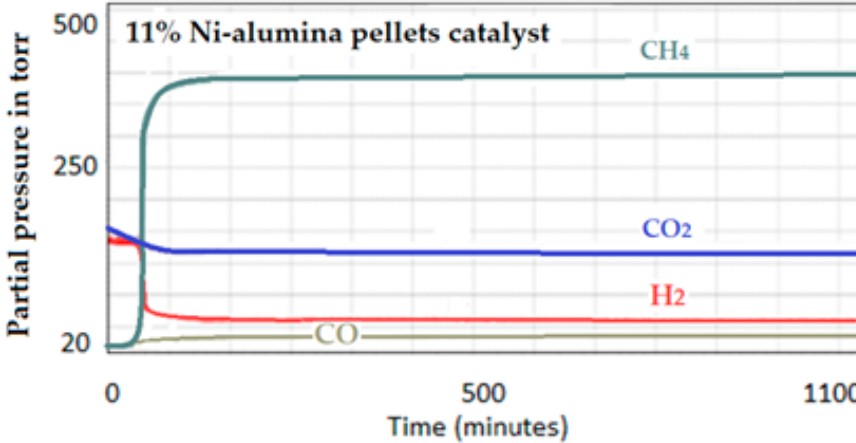

**Figure 24.** Intensities of the produced signals by the 11 wt.% Ni-supported alumina pellets catalyst.

(C) Catalytic test on the 4 wt.% Ni-supported alumina trilobe catalyst: Estimation of the performance for the 4 wt.% Ni-supported alumina trilobed catalyst. The activity spectrum for the catalyst up to 1000 min of reaction at 15 bar is presented in Figure 25. Same condition as in the former case. According to the performance estimation as was done on the previous case and dispersion of 25.5%, the activity of this catalyst to produce $CH_4$ is 983 ÷ 0.015 = 64,200.

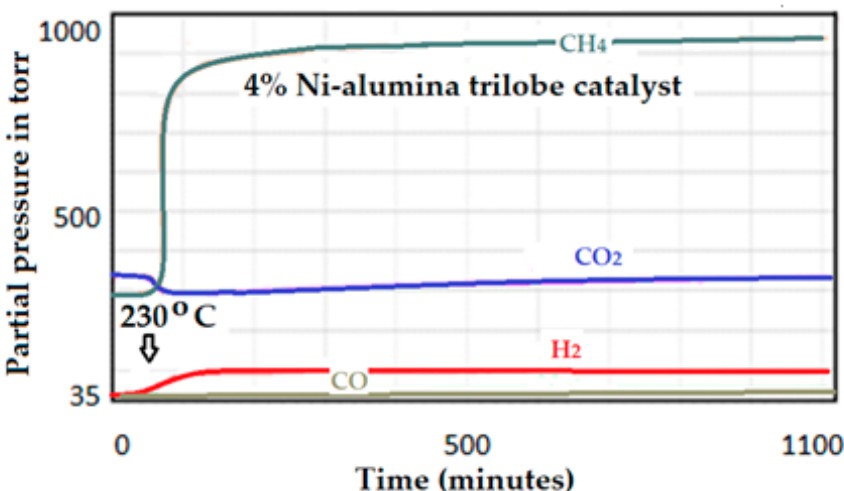

**Figure 25.** Intensities of the produced signals by the 4 wt.% Ni-supported alumina trilobe catalyst.

It can be concluded from these two results that the 4 wt.% Ni catalyst is about 25 times more active than the 13 wt.% CuO catalyst. This difference in performance could be due to several reasons: the low activity of the copper for this kind of reaction on one hand and the low surface area of the support compared to those of the alumina trilobe for the Ni catalysts. A higher surface area of the support yields higher dispersion of the active species, and hence, higher activity.

The produced signal by the 14.5 wt.% Ni-supported alumina trilobe catalyst is included in Figure 26. The catalytic test was carried out under the same conditions as previously described. The activity of the catalyst to produce $CH_4$ is determined as was also described in the previous tests.

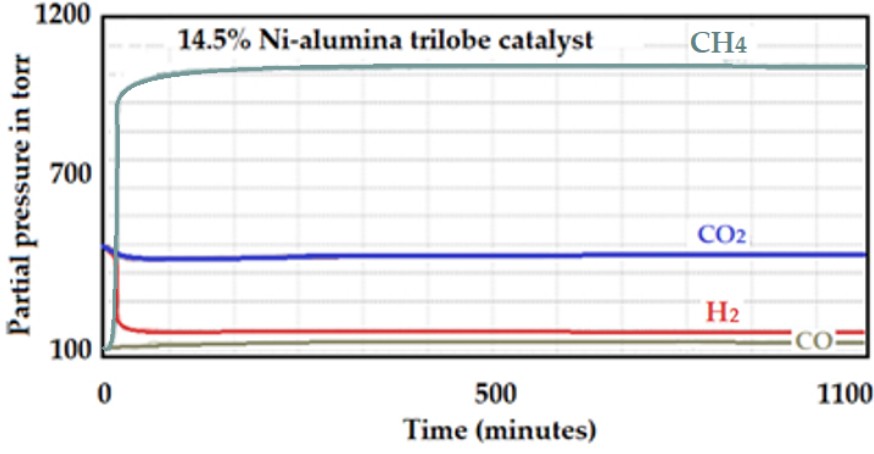

**Figure 26.** Intensities of the produced signals by the 14.5 wt.% Ni-supported alumina trilobe catalyst.

Dispersion for this catalyst was 5%.
The intensity of $CH_4$ produced was 986 and the total Ni on the surface was 0.004255.
Activity: 986 ÷ 0.004255 = 231,727.

### 3.1.3. Characterization of Catalysts upon Deactivation

TPO. At the end of the reaction, the catalyst was swept by an inert carrier gas, helium, and the reactor temperature was brought to room temperature. The first analysis was to perform a TPO to determine the amount of carbon that can be produced as a coproduct such as graphite or could also be due to some cracking in certain reactions. A flow of 100 $cm^3$/min of a 10% $O_2$ balance helium passed through the reactor at room temperature.

After signal stabilization, the reactor temperature was brought to 700 °C at a rate of 10 °C/min. The production of $CO_2$ and CO is mainly due to the oxidation of the graphite as a coproduct of the Sabatier reaction.

For the TPO analysis, the mass spectrometer can only be used as the TCD is unable to differentiate between the products, as in this case a mixture of CO and $CO_2$ is produced.

A mass spectrum corresponding to the oxidation of graphite by $O_2$ to produce $CO_2$ and CO for the used 13% Cu-alumina commercial catalyst is presented in Figure 27. The CO, in this case, could be due to the $CO_2$ signal and not to CO itself, as $CO_2$ shows a secondary signal at 28 *m/z* which is about 80% of the 44-signal intensity.

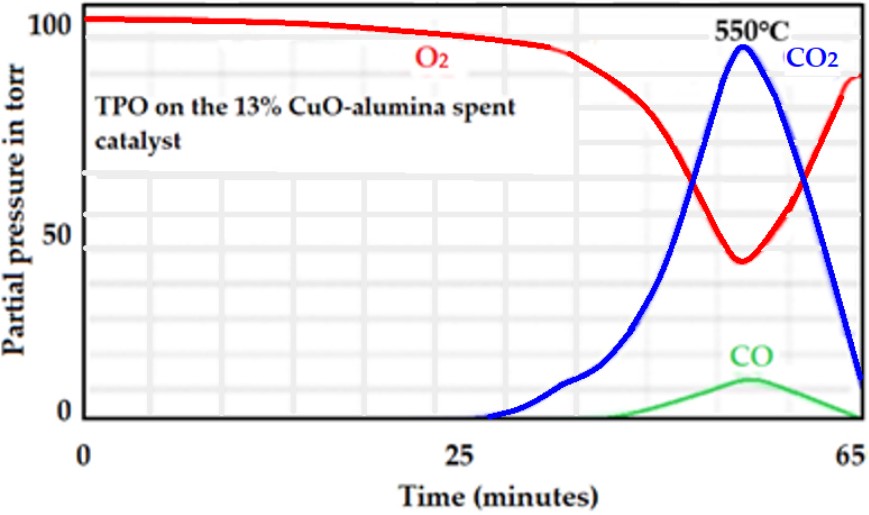

**Figure 27.** MS spectrum corresponding to the oxidation of the used 13 wt.% Cu-supported alumina catalyst.

The TPO spectrum and the production of $CO_2$ and CO produced by the used 11wt.% Ni-supported alumina pellets catalyst is presented in Figure 28. It is to be noted that the Ni-pellets catalyst produced much less $CO_2$ than the Cu-supported catalyst for the same reaction conditions.

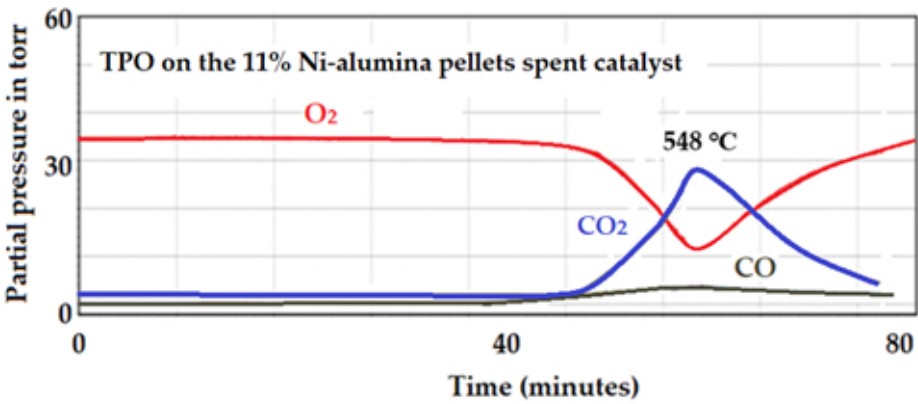

**Figure 28.** MS spectrum corresponding to the oxidation of the used 11 wt.% Ni-supported alumina pellets catalyst.

The spectrum of TPO on the 4 wt.% Ni-supported alumina trilobe catalyst is presented in Figure 29. In this case, this catalyst produced almost the same amount of $CO_2$ but also showed much higher activity to produce $CH_4$ than the two previous catalysts.

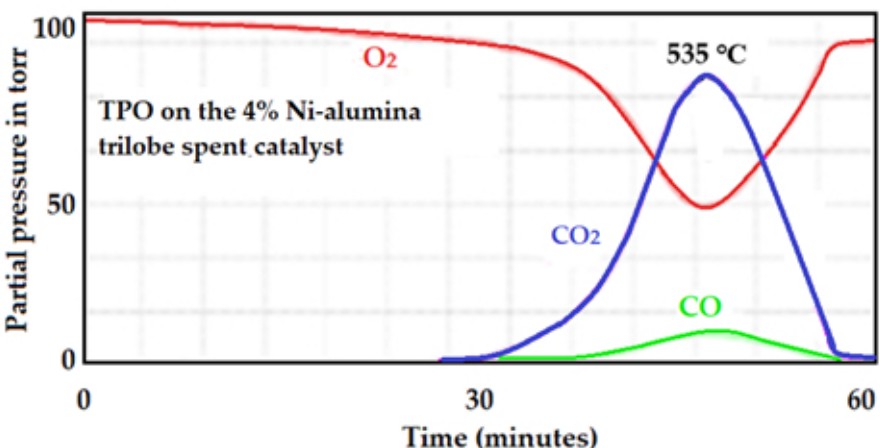

**Figure 29.** MS spectrum corresponding to the oxidation of the used 4 wt.% Ni-supported alumina trilobed spent catalyst.

Pulse. After TPO analyses, all catalysts were reduced with $H_2$ up to 550 °C to ensure full recovery of the metal elements. In situ CO pulse chemisorption was performed again at 35 to 40 °C to determine dispersion after the reaction. Differences in dispersion before and after the reaction would be indicative of sintering under the reaction conditions.

The $N_2O$ consumption by the Cu-supported alumina catalyst after the reaction is summarized in Figure 30. No $N_2$ was produced, which indicates that this catalyst was destroyed and sintered under the reaction conditions. The Cu active particles sintered to reduce dispersion as low as 2.5% (2.5 versus 29%) for the fresh catalyst.

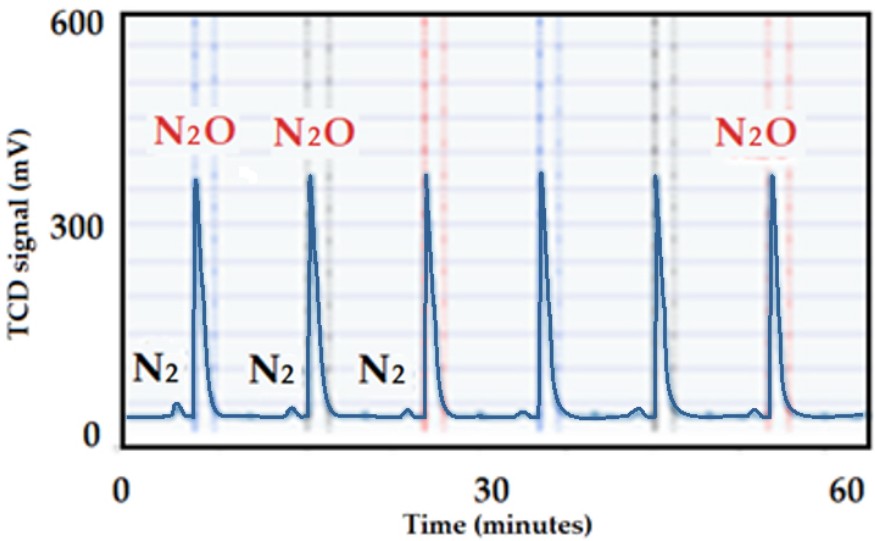

**Figure 30.** $N_2O$ chemisorption on the 13 wt.% Cu-supported alumina spent catalyst.

The CO pulse chemisorption was also in situ performed on the aged Ni-supported catalysts. The CO pulse chemisorption profile on the 11% Ni-alumina pellets catalyst after reaction is included in Figure 31. This analysis showed a result of 2.8% dispersion versus 3.3% dispersion before the reaction; this tiny difference in dispersion indicates that the catalyst was not highly affected by the reaction conditions.

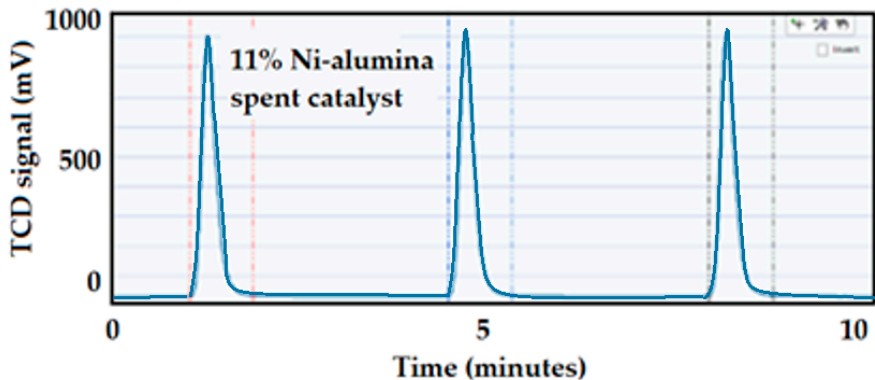

**Figure 31.** CO pulse chemisorption on the 11 wt.% Ni-supported alumina pellets spent catalyst.

The CO chemisorption on the 4 wt.% Ni-supported alumina trilobe catalyst after 20 h of reaction at 15 bar and 500 °C is included in Figure 32. The catalyst did not seem to be very affected by the reaction conditions as it showed a slightly lower dispersion value (35 versus 43%) for the fresh catalyst.

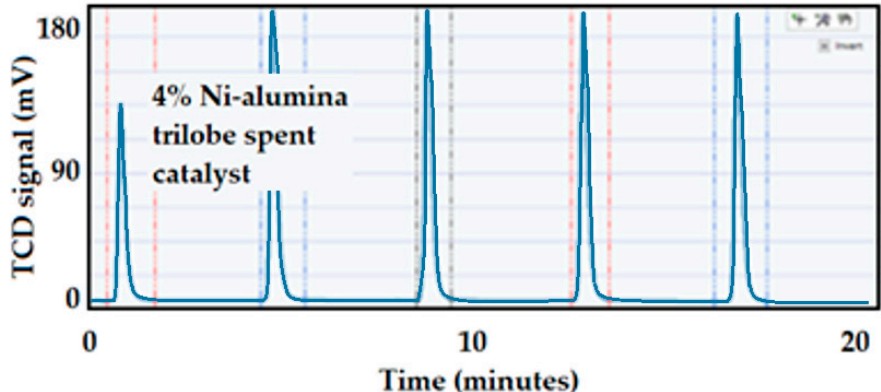

**Figure 32.** CO pulse chemisorption on the 4 wt.% Ni-supported alumina trilobe spent catalyst.

It can be concluded from the last test that in situ pulse chemisorption was necessary to determine the causes of catalyst deactivation.

Finally, it has been observed that the Ni-supported catalysts showed higher catalytic performance than the Cu-supported alumina commercial catalyst to produce $CH_4$. It should be noted that the higher the Ni content in the catalyst the higher the activity in producing $CH_4$ (14.5 wt.% Ni-trilobe < 11 wt.% Ni-alumina pellets < 4 wt.% Ni-trilobe < 13 wt.% CuO-alumina). The 4 wt.% Ni-alumina trilobe catalyst showed the highest dispersion and was not greatly affected by the reaction condition for at least 20 h at 15 bar, while the copper catalyst used for this study did not seem to be an active catalyst for the Sabatier reaction [83] that could be related to the large loading in copper. Possibly a lower loading in copper would yield higher dispersion, and therefore, possibly higher catalytic performance.

## 4. Summary, Conclusions, and Future Perspectives

Any catalyst, before being tested for a reaction, must undergo a specific protocol of characterization to simulate its behavior under high temperature and pressure reaction conditions. The first task of this study was to understand the importance of the textural and structural properties of the support used for the preparation of catalysts. The gas adsorption analysis completes knowledge of the texture of the support. The surface area provides information about the loading of metal particles that form a surface monolayer and become the active sites for the reaction. The porosity of the support plays an important role in the diffusion of the reactant molecules into the pores to access the active sites for

the reaction. The size of the pore will act like a sieve where only the reactant molecules of interest are allowed to diffuse and encounter the active particles that are normally located inside the pores. For an optimal diffusion of the reactant, the size of the pore should have an effective size that is normally five to six times larger than the reactant molecule diameter.

The study of the structure of the freshly prepared catalyst predicts the stability of the active species when the catalyst is subjected to severe reaction conditions, high temperature, and pressure, and the possibility to minimize sintering that is conducive to catalyst deactivation. Properties of the support, also make large contributions to the performance of the catalyst depending on the reaction, as an example, acidity for the hydrocarbon cracking reaction. Basic sites are also widely used in transesterification reactions (biodiesel production by using base catalysts, alkali metal hydroxides (NaOH/KOH), and alkali metals (NaOCH$_3$)) and many other reactions where a special active site is required.

The thermal programmed techniques are not only very attractive but also very accessible to many researchers. Techniques such as TPR are widely used to gain information about the catalyst. For example, the strength of interaction between the support and the active species is highly dependent on the preparation and calcination of the catalyst. It also serves to quantify the total amount of metal oxides available in the catalyst. The TPR profile can also yield information about the active particle size which is revealed by the mechanism of reduction.

Pulse chemisorption is also an important technique for the characterization of the catalyst. For example, the pulse technique is used as a titration method to quantify the number of active species available on the surface for the reaction for the reaction. Only the surface-active particles are accessible for the reaction and yield the activity of the catalyst. Hence, the importance of this technique, it can predict the activity and selectivity of the catalyst before the catalytic operation. This technique is also used to quantify the acidity of the catalyst as well as the strength of the acid sites at the reaction conditions. The total acidity of the catalyst is important; however, the distribution and strength of acid sites are more critical in catalysis.

After the characterization of the catalyst's property, testing becomes the second step in the life of the catalyst. The condition for the testing will depend on the reaction itself. For example, selecting the reaction conditions as well as the reactant molecules to produce the final desired product. In this study, a simple catalytic test was provided as an example, and a reduction of carbon dioxide was selected for the Sabatier reaction to complete the work for this review. Although GC is widely mentioned as a technique in the literature as the preferred analytical method, a mass spectrometer was instead selected for this study, as it allows a researcher to better follow the reaction steps and thus elucidate the formation of different products and the actual temperature at which reaction starts for each product.

Catalyst deactivation was also studied. Usually, there are two main causes of deactivation. During reactions at high temperatures and pressure, cracking can occur. Cracking is the production of carbon atoms as the result of the breakage of hydrocarbon molecules, or the formation of carbon species such as graphite for example as in the case of the Sabatier reaction. Carbon covers the active species preventing direct contact between the active particles and the reactant molecules. Hence, minimizing accessible catalytic sites or even completely deactivating the catalyst. Sintering of the active particles can also lead to deactivation. This occurs mainly at high temperatures when the support interaction with the active species is rather weak. The active particles, however, will start moving on the surface to collapse together forming larger particles and minimizing the active surface area where the reaction takes place. Characterization before the catalytic test is important, however, characterization after deactivation is crucial as well. A special condition for this study is that characterization must be done in situ without having to remove the catalyst from the reactor and expose it to an atmosphere where oxygen from the air can change the surface. One of the important issues mentioned in this study is the use of the new Micromeritics instrument, the ICCS, as was used in situ to characterize the catalyst before and after the reaction. TPO was used to identify the cause of deactivation as the deposition of carbon

onto the surface of the catalyst. Through TPO, the catalysts were reoxidized, producing both carbon monoxide and carbon dioxide. These two elements were monitored by the mass spectrometer that was connected to the exhaust of the reactor. The thermal conductivity detector (TCD) fails to differentiate between two elements, in this case, between CO and $CO_2$. All catalysts used in this study produced carbon during the reaction which contributes to the formation of carbon deposits, especially graphite, as one of the reaction products. The use of the Micromeritics FR-100 reactor for this study has the capability to remove the water produced by the Sabatier reaction at the reaction conditions (500 °C and 15 bar) from the primary product, $CH_4$ in this case.

The ICCS instrument allowed for titration of CO for the analysis of the Ni catalysts and the use of $N_2O$ for the Cu catalyst. It is to be noted that the ICCS includes a special column that can separate the $N_2$ from the excess of $N_2O$ without the need to use an external cryogenic trapping device. It has been concluded that the Ni-supported catalysts were not highly affected by the reaction conditions, especially for the low-loading catalyst, as dispersion did not reflect large changes as the Cu-supported catalyst showed. This effect could be because the Cu catalyst in this study contains a large amount of Cu (13 wt.%), which could result in agglomeration of the active particles. It could also be due to the low surface area of the catalyst support. While the Ni catalysts supported on the alumina trilobe enjoy a large surface area of the support and relatively low loading, that could lead to the high dispersion of the active particles, hence, higher activity for the Sabatier reaction.

An important conclusion from this study is that the use of a lower copper loading and larger surface area support could yield a more productive catalyst for the Sabatier reaction. Although it is widely mentioned in the literature that the use of copper as an active element for the Sabatier reaction is capable of producing not only $CH_4$ but also larger hydrocarbon molecules and some alcohol products.

Finally, for the $CO_2$ methanation Ni based catalysts results to be very effective and therefore are the most efficient and active catalytic system together with alumina. Even when Ru catalysts, or other precious metals, present better performances, Ru is about 120 times more expensive than Ni. Nickel catalysts have a short lifetime, because of carbon deposition which blocks pores and consequently deactivates the catalyst. A rational design of Ni-based methanation catalysts with high catalytic performance at low temperatures, good redox properties, and better stability at reaction temperatures could lead to a better option for industrial applications of $CO_2$ hydrogenation to methane. Knowing and understanding the stages that control the synthesis of a catalyst is crucial for its good design. This is the idea that has been tried to convey in this review work and that will be necessary to continue investigating in greater depth.

**Author Contributions:** All authors contributed to researching data for the article and writing and reviewing/editing the manuscript before submission. All authors have read and agreed to the published version of the manuscript.

**Funding:** A.G. is grateful for financial support from the Spanish Ministry of Science and Innovation (MCIN/AEI/10.13039/501100011033) through project PID2020-112656RB-C21.

**Institutional Review Board Statement:** Not applicable.

**Informed Consent Statement:** Not applicable.

**Data Availability Statement:** Not applicable.

**Acknowledgments:** A.G. is grateful for financial support from the Spanish Ministry of Science and Innovation (MCIN/AEI/10.13039/501100011033) through project PID2020-112656RB-C21. The authors are grateful to Micromeritics Instrument Corporation for the support.

**Conflicts of Interest:** The authors declare no competing interests.

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
