# Peer review of "On the Genesis of a Catalyst: A Brief Review with an Experimental Case Study"

_2673-4117, doi:10.3390/eng4030136_

Round 1
Reviewer 1 Report
Comments for Authors are reported in the attached document.

Author Response
Q1) Some typos are present in lines 30,346, 366, 375, 585.
Reply: The manuscript has been revised from these comments.
Q2) Please use superscript and subscript in lines 502 (CaCO3), 598 (H2), 833 (cm3/min), 978 (N2).
Reply: The manuscript has been revised from these comments.
Q3) I suggest reporting equations using a bigger size.
Reply: The manuscript has been revised from these comments.
Q4) I think that the authors made a detailed discussion about the various aspects concerning the synthesis and characterization of catalysts. However, more attention should be given to the images since some of them (e.g. Figs. 11, 13, 14) can be improved to better understand the concepts expressed in the manuscript
Reply: The manuscript has been revised from these comments.
Reviewer 2 Report
I have carefully reviewed the manuscript titled "Synthesis, Characterization, and Catalytic Application."
Overall, I find the work to be quite valuable in presenting the significance of heterogeneous catalysts in various fields such as CCS, renewable energy, water electrolysis for P2G, and the synthesis of green fuels. I am inclined to recommend acceptance of the manuscript with some revisions and significant structural modifications. Here are my specific comments:
General Comments:
i. It is recommended to expand the introduction section to include not only metal oxides but also metal Nitrides, Hydrides, and Carbides, and their roles in CCU or CO2 valorization processes.
ii. The synthesis methods discussed in the manuscript are conventional. It would be beneficial to explore more advanced techniques, focusing on industrial batch yield rates.
iii. On page 5, when referring to Eq. 2, please provide a detailed explanation of the unit of τ (mean crystal size). Additionally, clarify the units for FWHM and incident angle, expressed in radians.
iv. Figure 1 shows XRD patterns of TiO2 calcined at varying temperatures, but the physical significance of the peaks and what conclusions can be drawn from these patterns are not discussed.
v. On page 6, Figure 2 displays a typical XPS spectra, but it is not specified for which compounds or elements. Please explain how to interpret the spectra and what conclusions can be drawn.
vi. On page 15, in Equations (7) and (8), the symbols used should be defined in detail.
Synthesis, Characterization, and Catalytic Application:
i. Clarify whether the support materials, alumina trilobe, and alumina pellet, were synthesized or obtained commercially. Provide details about the products and mention the specific surface area of the support layers.
ii. On page 17, line 579, specify the type of reactor used for the synthesis process.
iii. On page 17, line 596, while the synthesis methods for the Ni infiltrate catalyst are discussed, there is no information about the source of the Cu for the Cu-supported catalysts. Please explain this.
iv. Chemical Adsorption, Chemisorption, or Selective Adsorption Technique:
v. For Figure 19, please rewrite the captions for each graph of CO pulse chemisorption for better clarity.
vi. In Figure 20, the TPR results suggest the formation of Ni-Al spinel phases. To confirm this, it would be beneficial to investigate the crystallite phase of the reduced Ni-Al catalysts using XRD techniques.
vii. On page 20, lines 699-703, consider revising the sentences for clarity. Also, clarify that the calcination process involves high-temperature thermal treatment under ambient conditions in the presence of air/O2.
Additionally, It is recommended that the author investigate the phase and microstructure, conduct elemental analysis of the synthesized catalysts, and discuss the results.
Addressing these points will enhance the quality and comprehensibility of your manuscript. I look forward to reviewing the revised version.
NA
Author Response
Q1) It is recommended to expand the introduction section to include not only metal oxides but also metal Nitrides, Hydrides, and Carbides, and their roles in CCU or CO2 valorization processes.
Reply: The Introduction Section has been revised from this comment. The title of the work has been also revised and modified.
Q2) The synthesis methods discussed in the manuscript are conventional. It would be beneficial to explore more advanced techniques, focusing on industrial batch yield rates.
Reply: We agree with the reviewer's comment that there is a great diversity of methods for preparing bulk and supported catalysts. We have selected the most commonly used methods both in research laboratories and in large-scale industrial processes. Specifically, in this work, we have selected precipitation methods, impregnation method, sol-gel method, and chemical deposition method, in order to provide certain information for the student who begins his studies in this field or for researchers whose field of experience is not heterogeneous catalysis. The objective remains to disperse the catalytic active phase in a material that acts as an inert support.
Q3) On page 5, when referring to Eq. 2, please provide a detailed explanation of the unit of τ (mean crystal size). Additionally, clarify the units for FWHM and incident angle, expressed in radians.
Reply: This information has been included in the revised version of the manuscript.
Q4) Figure 1 shows XRD patterns of TiO2 calcined at varying temperatures, but the physical significance of the peaks and what conclusions can be drawn from these patterns are not discussed.
Reply: More information has been included in the revised version of the manuscript.
Q5) On page 6, Figure 2 displays a typical XPS spectra, but it is not specified for which compounds or elements. Please explain how to interpret the spectra and what conclusions can be drawn.
Reply: More information has been included in the revised version of the manuscript.
Q6) On page 15, in Equations (7) and (8), the symbols used should be defined in detail.
Reply: This information has been included in the revised version of the manuscript.
Q7) Clarify whether the support materials, alumina trilobe, and alumina pellet, were synthesized or obtained commercially. Provide details about the products and mention the specific surface area of the support layers.
Reply: The catalytic supports used with different textural properties are commercial. This sentence has been modified in the revised version of the manuscript.
Q8) On page 17, line 579, specify the type of reactor used for the synthesis process.
Reply: The calcination of the catalytic precursor was carried out in the same flow bed reactor that is used for the catalytic tests. This sentence has been modified in the revised version of the manuscript.
Q9) On page 17, line 596, while the synthesis methods for the Ni infiltrate catalyst are discussed, there is no information about the source of the Cu for the Cu-supported catalysts. Please explain this.
Reply: We are considered a commercial copper catalyst (13 wt.% Cu-alumina) for comparison with the prepared Ni catalysts. Therefore, we have not carried out the synthesis of this catalyst.
Q10) For Figure 19, please rewrite the captions for each graph of CO pulse chemisorption for better clarity.
Reply: The caption of Figure 19 has been rewritten.
Q11) In Figure 20, the TPR results suggest the formation of Ni-Al spinel phases. To confirm this, it would be beneficial to investigate the crystallite phase of the reduced Ni-Al catalysts using XRD techniques.
Reply: We agree with the reviewer, but unfortunately these studies are not available because the catalytic tests were carried out at the Micromeritics company facilities. In its facilities there is no availability XRD apparatus to carry out these characterizations.
Q12) On page 20, lines 699-703, consider revising the sentences for clarity. Also, clarify that the calcination process involves high-temperature thermal treatment under ambient conditions in the presence of air/O2.
Reply: The sentence has been clarify in the revised version of the manuscript.
Q13) Additionally, It is recommended that the author investigate the phase and microstructure, conduct elemental analysis of the synthesized catalysts, and discuss the results.
Reply: We agree with the reviewer, but unfortunately these studies are not available because the catalytic tests were carried out at the Micromeritics company facilities. In its facilities there is no availability to carry out these characterizations.
Reviewer 3 Report
Novelty is not appropriate and should be improved. It should effectively convey the main novelty/contributions of the work.
Abstract is the summary of the paper presenting the main findings of the review conducted and not just a table of content.
Author Response
Q1) Novelty is not appropriate and should be improved. It should effectively convey the main novelty/contributions of the work.
Reply: The novelty of the work has been reviewed and highlighted in various parts of the work.
Q2) Abstract is the summary of the paper presenting the main findings of the review conducted and not just a table of content.
Reply: The abstract of the work has been revised taking this comment into account. In any case, a part of the work is a review of catalyst characterization techniques, that is why it is included as a table of contents.
Round 2
Reviewer 2 Report
The feedback received in response to the comments made on the initial manuscript has been carefully considered and found to be satisfactory.
I am inclined to approve the revised version of the manuscript in its present state.
The quality of the English language in the manuscript is deemed to be satisfactory.